# Global mycorrhizal plant distribution linked to terrestrial carbon stocks

Nadejda A. Soudzilovskaia [1]*, Peter M. van Bodegom [1,10], César Terrer [2,3,10], Maarten van't Zelfde[1], Ian McCallum[4,10], M. Luke McCormack [5], Joshua B. Fisher[6,7], Mark C. Brundrett [8], Nuno César de Sá [1] & Leho Tedersoo[9,10]

Vegetation impacts on ecosystem functioning are mediated by mycorrhizas, plant–fungal associations formed by most plant species. Ecosystems dominated by distinct mycorrhizal types differ strongly in their biogeochemistry. Quantitative analyses of mycorrhizal impacts on ecosystem functioning are hindered by the scarcity of information on mycorrhizal distributions. Here we present global, high-resolution maps of vegetation biomass distribution by dominant mycorrhizal associations. Arbuscular, ectomycorrhizal, and ericoid mycorrhizal vegetation store, respectively, $241 \pm 15$, $100 \pm 17$, and $7 \pm 1.8$ GT carbon in aboveground biomass, whereas non-mycorrhizal vegetation stores $29 \pm 5.5$ GT carbon. Soil carbon stocks in both topsoil and subsoil are positively related to the community-level biomass fraction of ectomycorrhizal plants, though the strength of this relationship varies across biomes. We show that human-induced transformations of Earth's ecosystems have reduced ectomycorrhizal vegetation, with potential ramifications to terrestrial carbon stocks. Our work provides a benchmark for spatially explicit and globally quantitative assessments of mycorrhizal impacts on ecosystem functioning and biogeochemical cycling.

[1] Institute of Environmental Sciences, Leiden University, 2333 CC Leiden, the Netherlands. [2] Institut de Ciència i Tecnologia Ambientals (ICTA) Universitat Autonoma de Barcelona, Barcelona, Spain. [3] Physical and Life Sciences Directorate, Lawrence Livermore National Laboratory, Livermore, CA 94550, USA. [4] Ecosystems Services and Management Program, International Institute for Applied Systems Analysis, Schlossplatz 1, A-2361 Laxenburg, Austria. [5] Center for Tree Science, The Morton Arboretum, 4100 Illinois Route 53, Lisle, IL 60532, USA. [6] Jet Propulsion Laboratory, California Institute of Technology, 4800 Oak Grove Dr., Pasadena, CA 91109, USA. [7] Joint Institute for Regional Earth System Science and Engineering, University of California, Los Angeles, CA, USA. [8] School of Biological Sciences, Faculty of Science, University of Western Australia, Crawley 6009 WA, Australia. [9] Natural History Museum and Institute of Ecology and Earth Sciences, University of Tartu, 14a Ravila, 50411 Tartu, Estonia. [10]These authors contributed equally: Peter M. van Bodegom, César Terrer, Ian McCallum, Leho Tedersoo *email: n.a.soudzilovskaia@cml.leidenuniv.nl

Mycorrhizas are mutualistic relationships between plants and fungi, in which fungi supply plants with nutrients and plants provide carbon to fungi[1]. Among mycorrhizal types, arbuscular mycorrhiza (AM), ectomycorrhiza (EcM) and ericoid mycorrhiza (ErM) are geographically the most widespread, colonizing over 85% of vascular plants across vegetated terrestrial biomes[1–4]. Due to the facilitation of plant nutrient acquisition[1] and the large biomass of fungal networks in soil[5], the presence and type of mycorrhiza are among the key determinants of ecosystem functioning[6–9] and biogeochemical cycling[10–13]. Thus, the types of mycorrhizal associations present likely also affect the global distribution of soil carbon stocks. There is growing evidence that ecosystems dominated by EcM and ErM vegetation exhibit higher topsoil carbon to nitrogen ratios (C/N) compared with ecosystems dominated by AM plants[11,12,14,15], although in temperate forests the pattern may be reversed in deeper soil layers[16]. The mechanisms driving these differences are heavily debated in current literature, with distinct physiological traits of mycorrhizal fungi most likely playing a critical role[16–20].

Although it can be argued that high abundance of EcM plants is a consequence rather than a driver of high soil C stocks, a large body of recent findings provides evidence that EcM symbionts may be the key drivers of topsoil carbon accumulation through two interacting mechanisms. First, EcM fungi produce greater biomass of more recalcitrant mycelium compared to AM fungi[5]. Second, while EcM fungi are more efficient in taking up N in N-poor soils than AM fungi or roots[10,21], EcM fungi immobilize most of the N in their own biomass. This suppresses saprotrophic decomposition process[22] and reinforces the competitive advantage of EcM and ErM plants via enhanced N limitation[22,23].

A full understanding of global carbon and nitrogen stocks requires quantitative models on the distribution of mycorrhizal types in ecosystems[18]. Despite the existence of regional maps of current[24,25] and past[26] mycorrhizal vegetation, and on the distribution of mycorrhizal fungal species[27,28] we still lack global information on the distribution of biomass of mycorrhizal plants, which is a much better proxy for mycorrhizal impacts on ecosystem functioning than the biodiversity of mycorrhizal symbionts. While current terrestrial biosphere models simulate feedbacks between the carbon cycle and vegetation distribution[29], most models ignore mycorrhizal types and their effects on nutrient cycling. Integration of such information is expected to provide a more realistic simulation of carbon and nutrient fluxes associated with plant nutrition[17,18,21] and soil carbon cycles[15,21]. Quantitative models of the distribution of mycorrhizal vegetation constitute an important missing link between the known effects of mycorrhizas in biogeochemical cycles and their global impacts[30].

Human activities such as forest logging, urbanization and agricultural practices have altered 50–75% of the Earth's terrestrial ecosystems[31], transforming areas with previously natural EcM and ErM vegetation into AM and non-mycorrhizal (NM) vegetation. However, the impact of anthropogenic land use shifts on biogeochemical cycles associated with mycorrhiza have remained poorly known due to the lack of appropriate spatial information.

Based on a comprehensive quantitative evaluation of plant-mycorrhizal associations and the distribution of vascular plant species across biomes and continents, we assembled high-resolution digital maps of the global distribution of biomass fractions of AM, EcM, ErM and NM plants. Building on these maps, we assessed: (i) the amount of aboveground biomass carbon currently stored in each type of mycorrhizal vegetation; (ii) the impact of conversion of natural ecosystems to croplands on the distribution of mycorrhizal types globally; and (iii) the relationships between relative abundance of AM and EcM plants in

an ecosystem and soil carbon content in topsoil (0–20 cm), medium (20–60) and deep (60–100 cm) subsoil layers.

## Results and Discussion

**Assembly of mycorrhizal vegetation maps.** To generate global maps of mycorrhizal vegetation, we estimated biomass fractions of AM, EcM, ErM and NM plants within each combination of continent × ecoregion × land cover type. Supplementary Fig. 1 illustrates the data assembly processes for the maps. Ecoregions follow Bailey[32] (Supplementary Data 1), and land cover types were retrieved from the ESA CCI land cover map[33], which specifies cover and biomass fractions of trees, shrubs and herbaceous plants (Supplementary Data 2). For each combination, we determined the dominant species or group of species from 1568 vegetation surveys (Supplementary Data 3). For these species, we determined mycorrhizal type using the FungalRoot database v1.0[34] (see Supplementary Data 4 for data sources). Integrating these data, we obtained mycorrhizal plant biomass fractions of AM, EcM, ErM and NM plants for each combination of Bailey ecoregion, continent, and land cover type (Supplementary Data 5 and 6). These fractions were overlain on a global grid.

Our maps (Fig. 1) provide quantitative estimates of the distribution of aboveground biomass fractions among AM, EcM and ErM plants within areal units of 10 arcmin. The use of a detailed map of ecoregions[32] provides much greater resolution compared with the biome-based patterns of mycorrhizal distributions reported by Read[3] > 25 years ago, whereas the land cover map[33] enabled us to provide accurate spatial positioning of ecosystem boundaries based on satellite-derived data, explicitly taking into account human-driven transformations of vegetation.

We validated the map data using four independent datasets: (i) forest biomass structure for Eurasia[35], (ii) a global dataset of forest biomass structure used for an analysis of mycorrhizal impacts on carbon vs nitrogen dynamics[19], (iii) estimates of mycorrhizal associations in the USA based on satellite remote sensing[36], and (iv) West Australian map of mycorrhizal root abundance[24] (Supplementary Fig. 2). This validation revealed that the vast majority of the data (87% of the AM data points and 89% of the EcM data points) deviate by < 25% from the measurements[19,35,36], when excluding ESA land use classes comprising poorly resolved combinations (i.e. mixed classes of land cover types[33], such as "Tree cover, broadleaved, evergreen, closed to open (>15%)", see Methods for details) that were difficult to couple to our classification scheme. The relationship between the validation data and our estimates is shown in Supplementary Fig. 3.

Our maps of mycorrhizal vegetation were assembled based on multiple published datasets, using a number of conversion factors to obtain per pixel values of mycorrhizal plants biomass fractions. These conversions as well as the fact that the plant species distribution data (Supplementary Data 4) originates from multiple sources constitute important uncertainty sources in our dataset. We examined the uncertainty of our maps based on uncertainties of tree, shrubs and herbaceous plant fractions within the land cover types[37], and the number of data sources used to assess mycorrhizal fractions of plant biomass within each combination of Bailey ecoregion × continent; see Methods for details. Supplementary Fig. 4 shows spatial distribution of uncertainties. The mean uncertainties of AM, EcM, ER and NM maps are 19.6, 17.6, 14.6 and 15.0% at the 90% confidence interval. Overall, tropical areas have the highest uncertainties of the mycorrhizal fraction data, reaching 50% (AM) in the Amazon region. Therefore, our maps should be used with caution for these areas. Future sampling efforts of mycorrhizal vegetation distribution should be more focussed on tropical areas of Asia, Africa and South America.

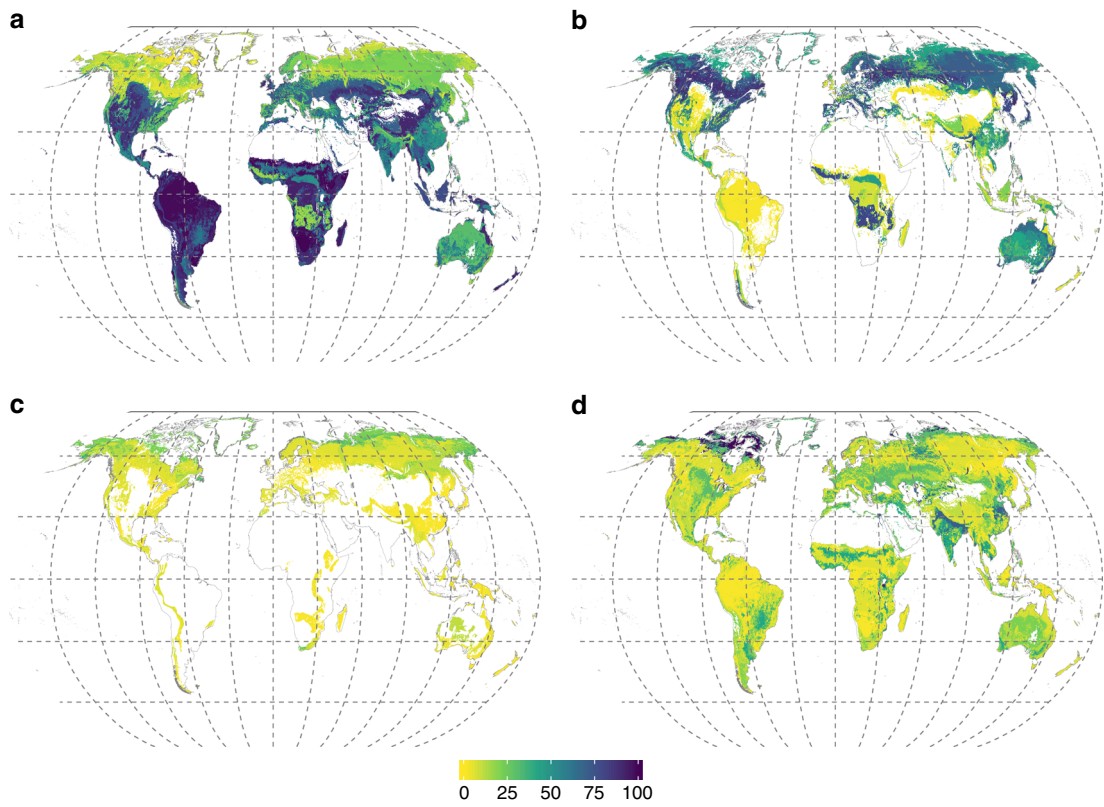

**Fig. 1** Percentage of aboveground plant biomass of mycorrhizal vegetation. **a** Arbuscular mycorrhizal plants, **b** ectomycorrhizal plants, **c** ericoid mycorrhizal plants, and **d** non-mycorrhizal plants. The map resolution is 10 arcmin. See Supplementary Fig. 4 for associated uncertainty values. Source data are provided as a Source Data file

**Mycorrhizal vegetation and aboveground carbon stocks**. By linking our maps of mycorrhizal vegetation to satellite observations of global aboveground biomass carbon[38], we estimated the amount of aboveground biomass carbon stored in arbuscular, ecto-, ericoid and non-mycorrhizal vegetation as 241 ± 15, 100 ± 17, 7 ± 1.8 and 29 ± 5.5 GT (mean values ± uncertainty at 90% confidence interval; Fig. 2). In this analysis, the data were scaled to a resolution of 15 arcmin to match biomass estimates[38]. Most of the aboveground carbon stock stored in arbuscular mycorrhizal vegetation is situated in tropical forests (Fig. 2a). Supplementary Table 1 shows per-biome distribution of the carbon stocks among mycorrhizal types.

**Impacts of land transformations on mycorrhizal vegetation**. Agricultural practices drive the replacement of natural vegetation by facultatively AM crops[1,39], which could also be de facto non-mycorrhizal due to destruction of hyphal networks by ploughing and excess fertilisation[40,41]. Using past vegetation estimates, Swaty et al.[26] showed that across conterminous USA, agriculture has reduced the relative abundance of ectomycorrhizal plants compared with other mycorrhizal types. However, global quantifications of agricultural impacts on distribution of mycorrhizas have not been possible until now. Based on the current land use data underlying our maps (Supplementary Data 4), we assessed mycorrhizal distributions on Earth in the absence of croplands. For each ecoregion-continent-land cover combination that contained croplands, we replaced current biomass fractions by estimates of per-grid cell biomass fractions in AM, EcM, ErM and NM plants that would be expected at these locations based on natural vegetation types (see Methods for details, and Supplementary Data 7–8 for data). Based on these data, we generated maps presenting potential natural distributions of biomass

fraction of AM, EcM, ErM and NM plants in a cropland-free world (Supplementary Figs. 6 and 7). The current biomass fractions of AM plants have increased in Europe, parts of Asia and North America, but declined in Africa, Asia (mostly India) and South America, coinciding with increase in non-mycorrhizal vegetation (Fig. 3, Supplementary Fig. 8). Our analysis suggests that EcM biomass has declined in all continents, primarily due to a replacement of natural forests by agricultural lands, whereas ErM biomass has remained unchanged.

**Biomass fractions of mycorrhizal types and soil C stocks**. Recent field research in the US temperate forests suggests that soil carbon content increases with increasing EcM abundance in topsoil layers; but, depending on forest type, this relationship may be reversed in deeper soil[16,42]. This is in agreement with the Microbial Efficiency-Matrix Stabilization hypothesis, which predicts that ecosystems with rapid decomposition, such as most AM-dominated forests[19], enhance soil organic matter (SOM) stabilization by accelerating the production and deposition of microbial residues[43–45].

To separate the effects of biome and mycorrhizal type on soil C on a global scale, we modelled the relationships between soil carbon content, biome type[46], and biomass fractions of AM and EcM plants. We did not analyse the relationship between ErM plant cover and soil C content, due to a small proportion of the ErM plant biomass in the majority of ecosystems.

We conducted separate analyses for the topsoil (uppermost 20 cm soil layer) and subsoil (20–60 and 60–100 cm soil layers), as obtained from the ISRIC-WISE Soil Property Databases, at a resolution of 30 arcsec[47]. The data sources used for these analyses are independent: the ecoregion classification, and hence mycorrhizal type distribution, does not account for edaphic parameters,

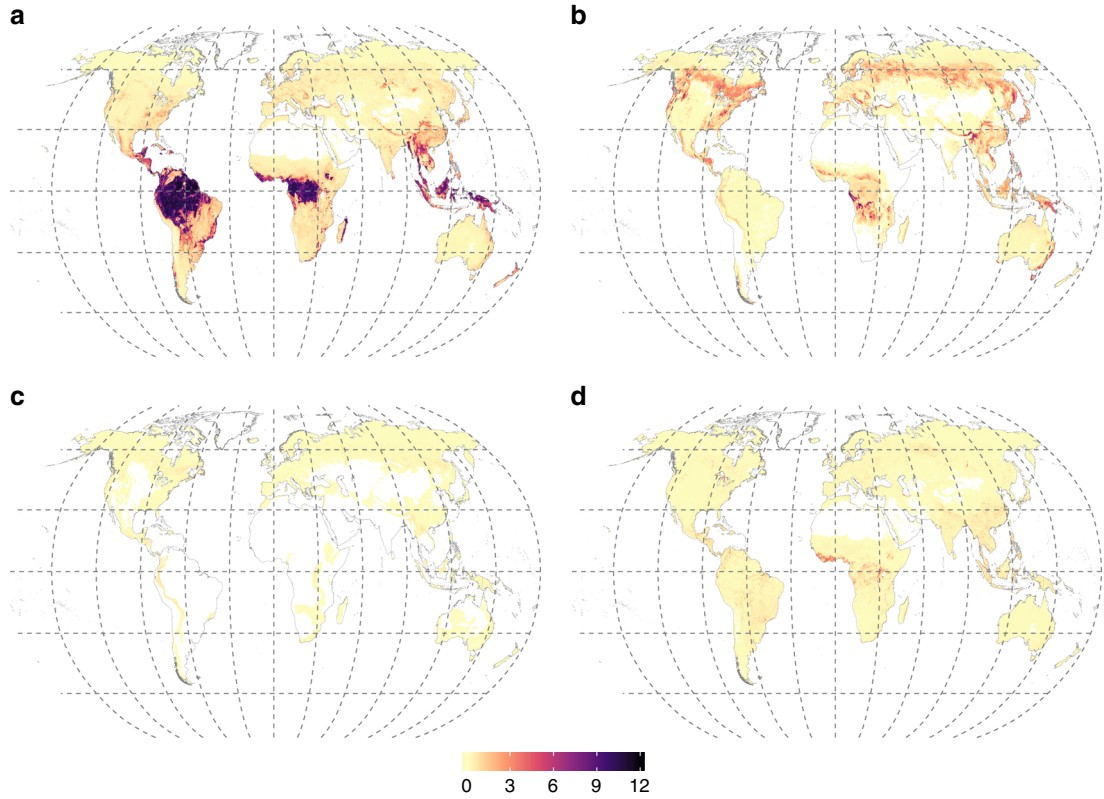

**Fig. 2** Amount of carbon stored in plant biomass in vegetation of different mycorrhizal types (Mt C per-grid cell of 15 arcmin). **a** Arbuscular mycorrhizal plants, **b** ectomycorrhizal plants, **c** ericoid mycorrhizal plants, **d** non-mycorrhizal plants. The amount of aboveground biomass carbon stored in arbuscular, ecto-, ericoid and non-mycorrhizal vegetation is $241 \pm 15$, $100 \pm 17$, $7 \pm 1.8$ and $29 \pm 5.5$ GT (mean values ± uncertainty at 90% confidence interval), respectively

whereas soil C data are unrelated to that of vegetation. Model comparisons were based on the Akaike information criterion (AIC). The relative importance of each predictor was examined using the Lindemann-Merenda-Gold (LMG) metric, providing the fraction of variance explained by each predictor, within the total variance explained by the model.

Our global analysis revealed a moderately strong positive relationship between the biomass fractions of EcM plants and both topsoil and subsoil carbon (Fig. 4, Supplementary Figs. 8 and 9). Consistent with the current paradigm[45], cf. ref. [11], our analysis revealed that biome is the main predictor of soil carbon stocks. Even so, the grid cell biomass fraction of EcM plants still accounted for one third of the explained variation in both topsoil carbon and in subsoil (Table 1). The interaction between biome and EcM biomass fractions was significant ($P < 0.001$) but only marginally important (LMG = 1%), suggesting that the increase in topsoil carbon along with an increase in EcM plant biomass is mostly independent from the environment (Table 1, Fig. 4, Supplementary Figs. 8, 9, Supplementary Table 2). The total aboveground carbon stock stored in the EcM plants was a relatively worse predictor of soil carbon stocks compared with the EcM relative biomass fraction, showing in all cases higher AIC and lower $R^2$. In contrast to EcM, AM biomass fractions per-grid cell showed negative but inconsistent relationships to soil carbon, with a contrasting positive trend in tundra (Fig. 4, Supplementary Fig. 10). The latter relationship explained only a small amount of variance in tundra soil carbon stocks (12% in top 0–20 cm, 1.7% in 20–60 cm layer and 0.2% in 60–100 cm layer), and could arise from less accurate data of soil C in tundra[47] due to local landscape heterogeneity and prevalence of facultative AM plants with no or low mycorrhizal colonization[48].

Our analysis of the relationships between mycorrhizal type and soil carbon is based on ancillary maps, which feature large uncertainties. Analyses of relationships between ISRIC-WISE predicted soil carbon and the original data that were used to generate the ISRIC soil map yield $R^2$-values in the range of 0.4–0.6[49,50]. This uncertainty adds ambiguity to our analysis, and reduces the reliability of quantitative estimates of the relationships between EcM plant biomass fractions and soil C. However, these uncertainties equally apply to the analysis of AM vs soil C as well as to that of the EcM vs soil C, due to the fact that the analysis is based on the same geographical data points. Therefore, we consider that the high uncertainty of the ISRIC-WISE soil data is unlikely to affect the qualitative nature of our conclusion that AM and EcM vegetation differently relate to soil C.

**Implications**. Our work provides the quantitative estimates of the global biomass distribution of arbuscular, ecto-, ericoid and non-mycorrhizal plants, accounting for human-induced transformation of habitats. Previous research has shed light onto the distribution patterns of mycorrhizal plant and fungal species richness[27,28], and onto low-resolution (1 arcdegree) distribution patterns of mycorrhizal trees[51]. In contrast, our maps directly reflect the global distribution of biomass fractions of mycorrhizal plants across all biomes and all main vegetation types. Availability of such data at high resolution of 10 arcmin provides an opportunity for multiple potential analyses aimed at unravelling mycorrhizal impacts on ecosystem functioning at large-geographical scales.

Our maps were derived at spatial resolutions allowing identification of the global patterns of mycorrhizal distributions and are most appropriate for global and large-geographical scale

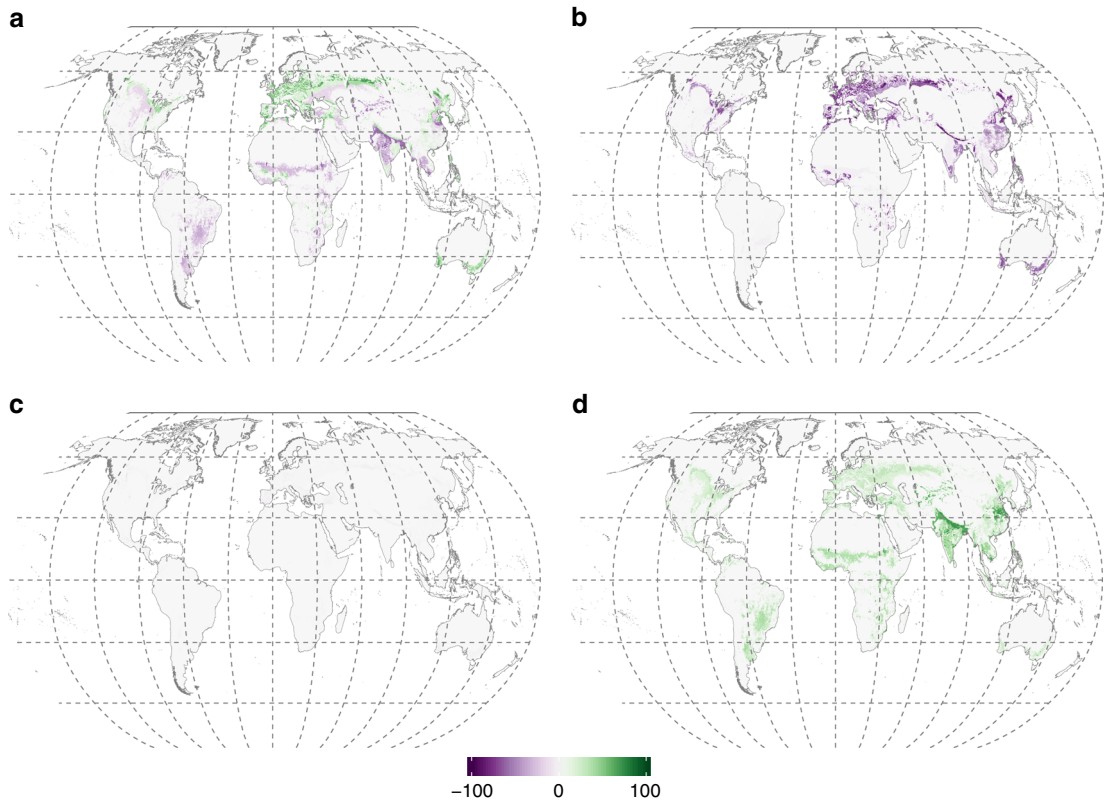

**Fig. 3** Changes in biomass fractions of mycorrhizal vegetation induced by crop cultivation and pastures. **a** Arbuscular mycorrhizal plants, **b** ectomycorrhizal plants, **c** ericoid mycorrhizal plants, **d** non-mycorrhizal plants. Purple colours indicate losses, green colours indicate gains. Uncertainties are shown in Supplementary Fig. 7. Source data are provided as a Source Data file

analyses of mycorrhizal impacts on ecosystem functioning and global drivers thereof. Recent estimates suggest that the total soil carbon loss due to agricultural practices accounts for 133 GT, with great acceleration of losses during the past 200 years[52]. Our analyses accounting for pre-agricultural patterns in EcM plant distribution point to large-scale losses of ectomycorrhizal vegetation, with potentially strong effect on the amount of C stored in soils. Analyses of agricultural impacts presented in this paper are based on the assumption that these impacts are limited to shifts in plant species composition, and do not encompass shifts in soil water and nutrient availability, which could affect activity of mycorrhizal fungi, depth distribution of mycorrhizas in the soil, and shifts among mycorrhizal fungal species composition due to the introduction of exotic species. While such simplifications are necessary for the analyses reported in this paper, they should be considered when interpreting our results. Furthermore, our analyses do not address other human impacts that can lead to shifts among AM and EcM vegetation, such as climate change, introduction of invasive species and nitrogen deposition. The latter is known to be an especially important driver of mycorrhizal vegetation shifts[42,53], as it negatively affects abundance of ectomycorrhizal fungi in soil[54]. Given that nitrogen deposition leads to replacement of ectomycorrhizal plants by arbuscular and non-mycorrhizal vegetation, it further enhances agricultural impacts on soil carbon losses.

The question whether increased domination of EcM plants in an ecosystem is associated with higher soil carbon content across both top- and subsoil is heavily debated[11,12,16,18–20,30]. However previous studies have been based on a limited number of observations[11] or regional-scale analyses[11,12,16,53,55]. Our analysis shows that across large geographical scales, higher cover of EcM vegetation is broadly associated with greater soil C stocks in both

topsoil and subsoil, while AM vegetation has more variable, weaker and mostly negative relationships. This analysis does not provide evidence of causality of this relationship as multiple environmental variables such as climate, soil nutrients, especially nitrogen availability, and soil texture may affect both soil carbon and mycorrhizal plant distributions. Nonetheless, our study establishes a quantitative framework to test the relation between the dominance of mycorrhizal types and soil C stocks. Complete and directional understanding of the complex nature of the hierarchy of environmental drivers controlling soil carbon patterns requires further detailed (experimental) investigations of the hierarchy of different predictors and importance of local edaphic variables.

Our estimates of the carbon stocks in AM and EcM aboveground biomass together with the quantitative relationships between soil carbon stocks and AM or EcM plant dominance in ecosystems provide qualitative insights into the global carbon cycle, highlighting the substantial role of mycorrhizas therein. Tropical forests, usually dominated by AM symbiosis (Fig. 2), contain 162 GT (44%) of global aboveground biomass[38], whereas the predominately EcM temperate and boreal forests altogether store only 21% of global aboveground biomass carbon[38], indicating that contribution of EcM vegetation to the aboveground biomass carbon is relatively small. In contrast, belowground carbon stocks are positively correlated to the proportion of EcM plant biomass, suggesting that mycorrhizal contribution to large carbon stocks in these regions occurs primarily through the carbon supply to belowground organs and mycorrhizal fungi, which is further emphasized with slowed decomposition processes[22]. Furthermore, our analyses revealed a relatively stronger relationship between soil C and EcM plant biomass fraction (%) than between soil C and the amount of carbon stored

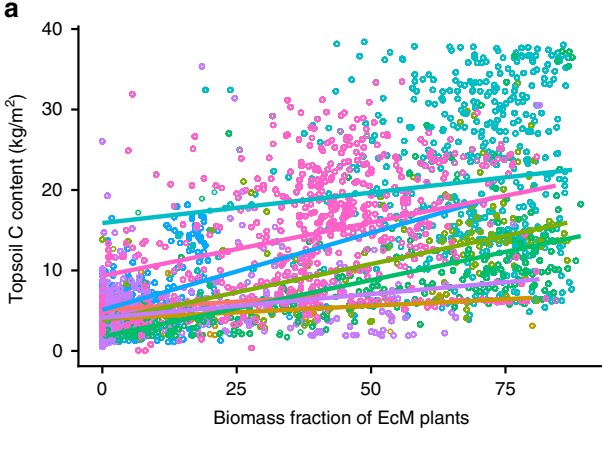

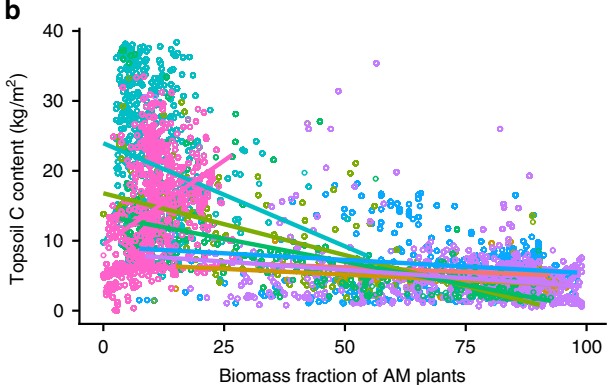

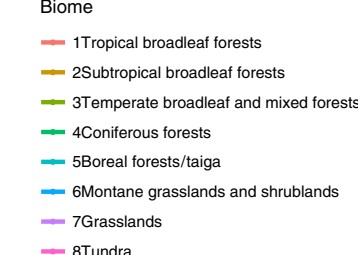

**Biome**

— 1 Tropical broadleaf forests
— 2 Subtropical broadleaf forests
— 3 Temperate broadleaf and mixed forests
— 4 Coniferous forests
— 5 Boreal forests/taiga
— 6 Montane grasslands and shrublands
— 7 Grasslands
— 8 Tundra

**Fig. 4** Quantitative relationships between topsoil (0–20 cm) C and biomass fraction of mycorrhizal vegetation in natural ecosystems. **a** EcM plants and **b** AM plants. The outcomes of individual models are presented in the Supplementary Table 2. Croplands were excluded from the analysis. Per-biome predictions are shown in different colours. Source data are provided as a Source Data file

in EcM plants. These findings suggest that belowground carbon allocation by plants through mycorrhizal pathways is not directly proportional to the aboveground plant biomass, supporting the view of the importance of mutualism-parasitism trade-off in ectomycorrhizal associations[56] across biomes.

Taken together, this study provides a benchmark for relating ecosystem processes to the functioning of distinct types of mycorrhizas on a global scale. So far, quantitative global information about mycorrhizal distribution was virtually absent despite the high demand for such data[12,13,30]. In spite of some uncertainty, our mycorrhizal distribution maps provide an essential source for systematic analyses of mycorrhizal biogeography and environmental drivers. Because our maps are based on field data, and not on a machine-learning model trained with environmental variables, they provide independent data for examining the relationships between mycorrhizal status and

ecosystem functioning, without introducing a circular reasoning caused by the use of common environmental variables. Inclusion of mycorrhizal distribution into vegetation models would provide a benchmark for testing hypotheses about mycorrhizal impacts on ecosystem functioning and related ecosystem services. Our maps enable quantifying relationships between mycorrhizal abundances in ecosystems as well as soil and vegetation carbon content in global-scale analyses of biogeochemical cycles. In particular, the results of our study suggest that restoration of native vegetation especially in abandoned agricultural and barren land may help alleviate anthropogenic soil carbon losses and ameliorate increases in atmospheric greenhouse gases.

## Methods

**Data used for construction of mycorrhizal vegetation maps.** To construct the maps of mycorrhizal biomass fraction distribution, we integrated (1) data concerning dominant plant species and their growth forms within each continent × ecoregions × land cover type combination, (2) data describing mycorrhizal type of these species, and (3) data estimating cover and biomass fractions of trees, shrubs and herbaceous plants within individual land cover types. Here we defined a species or a set of species with a similar mycorrhizal strategy and growth form as 'dominant' if it constituted > 20% of vegetation biomass. Supplementary Fig. 1 shows a flowchart of the map assembly process.

We selected the global ecoregion map of Bailey[32] with 98 ecoregions (Supplementary Data 1), provided by the Oak Ridge National Laboratory Distributed Active Archive Center[57] (spatial resolution 10 arcmin), as a basis for mapping of global-scale distribution of mycorrhizal types. This map was preferred over that of biomes provided by Olson and co-workers[46] and others because of higher level of detail for ecoregions and because the boundaries of ecoregions were more strongly related to the distribution of mycorrhizal types. Ecoregions spanning across multiple continents were considered separately for each continent. We used the continent division based upon the FAO Global Administrative Unit Layers (http://www.fao.org/geonetwork/srv/en/).

We determined the land cover types present in each ecoregion using a satellite observation-based map for the year 2015 generated by the European Space Agency[33]. This map includes 38 land cover categories such as croplands, urban areas, grasslands and forests of various types, with a spatial resolution of 300 m (Supplementary Data 2).

Based on vegetation surveys assigned to ecoregions (1568 data sources; Supplementary Data 3) we determined the dominant plant species and their growth form for each continent × ecoregion × land cover type combination. We used relative abundances of plant species averaged across different data points, as available across vegetation surveys, with equal weight to all observations.

We assigned mycorrhizal type (AM, EcM, ErM, NM) to each dominant species using the FungalRoot database[34]. Species with dual EcM-AM and AM-NM categories were allocated to both types equally (50% weight). Mycorrhizal status of species with no empirical records were extrapolated from congeneric and confamilial species. Therefore, all Diapensiaceae and Ericaceae species were considered ErM[58], except for *Enkianthus* (AM), *Arbuteae, Pyroleae, Monotropeae* and *Pterosporeae* (all subtypes of EcM). Because of multiple incorrect reports and alternative definitions for EcM, we took a conservative approach by considering plants to be EcM only when this was supported by multiple independent studies and the proportion of conflicting reports was < 50%[59]. Although most crop plants are able to form arbuscular mycorrhizas, intensive agricultural practices and breeding may lead to reduction or loss of mycorrhizal infection[40,41]. Therefore, rain fed and flooded croplands were considered to feature partly AM and partly NM vegetation (Supplementary Data 5, 7), unless data indicating dominance of NM vegetation was available. Crop species that belong to *Brassicaceae* family were considered NM.

Combining the prevailing dominant plant species, their growth form and mycorrhizal type, we estimated the biomass proportions of EcM, AM, ErM and NM vegetation in each ecoregion by continent by land cover combination (Supplementary Data 5 and 6). We considered that in forests with a sparse understorey, trees contribute 90–95% of the biomass and the understorey accounts for 5–10% of biomass[36,60–63]. In forests with a dense layer of shrubs, trees, shrubs and herbs/dwarf shrubs contribute 70 ± 15%, 20 ± 10% and 10 ± 5% to plant biomass[36,60–63], respectively. In shrublands, we consider shrubs and herbs to account for 90 ± 5% and 10 ± 5% of biomass, respectively[35,60–63]. We considered that woodlands harbour 30 ± 10% of biomass in trees, 30 ± 10% of biomass in shrubs, and the remaining biomass in herbaceous vegetation[35,60–63]. This resulted in biomass proportions of each mycorrhizal type in continent × ecoregion × land cover type (Supplementary Data 5, 6).

As we focused on the biomass of mycorrhizal plants and not on species diversity; we did not attempt to map the distribution of orchid mycorrhiza. Orchid species are never abundant in ecosystems in terms of biomass, and they are therefore unlikely to play an essential role in biogeochemical cycles at large regional scales.

**Table 1 Summary of generalized linear models (glm) predicting soil carbon stocks**

| Predicted variable | Model | $R^2$ | Predictor | P-value | LMG (%) |
|---|---|---|---|---|---|
| Topsoil C 0–20 cm | EcM + Biome + EcM × Biome | 0.53 | EcM | <0.001 | 42 |
| | | | Biome | <0.001 | 57 |
| | | | EcM × Biome | <0.001 | 1 |
| Subsoil C 20–60 cm | EcM + Biome + EcM × Biome | 0.38 | EcM | <0.001 | 39 |
| | | | Biome | <0.001 | 60 |
| | | | EcM × Biome | <0.001 | 1 |
| Subsoil C 60–100 cm | EcM + Biome + EcM × Biome | 0.33 | EcM | <0.001 | 35 |
| | | | Biome | <0.001 | 64 |
| | | | EcM × Biome | <0.001 | 1 |
| Topsoil C 0–20 cm | AM + Biome + AM × Biome | 0.54 | AM | <0.001 | 38 |
| | | | Biome | <0.001 | 56 |
| | | | AM × Biome | <0.001 | 6 |
| Subsoil C 20–60 cm | AM + Biome + AM × Biome | 0.33 | AM | <0.001 | 29 |
| | | | Biome | <0.001 | 67 |
| | | | AM × Biome | <0.001 | 2 |
| Subsoil C 60–100 cm | AM + Biome + AM × Biome | 0.32 | AM | <0.001 | 31 |
| | | | Biome | <0.001 | 67 |
| | | | AM × Biome | <0.001 | 2 |

Predictions are made for C at 0–20, 20–60, and 60–100 cm depth and are based on biome and fraction of EcM or AM plants in vegetation biomass. $R^2$—Cragg and Uhler's pseudo $R^2$. LMG—relative importance of individual predictors in a model examined through the Lindemann-Merenda-Gold metric. The LMG shows the percentage of variance explained by each of model predictors within the entire variance explained. The P values show the outcome of ANOVA type I models ($n = 78883$ in all models). Source data are provided as a Source Data file

**Assembly of raster maps of mycorrhizal vegetation**. We generated raster maps based on the proportional mycorrhizal type biomass data. We overlaid the raster map of Bailey ecoregions (10 arcmin resolution)[57] with the raster of ESA CCI land cover data (300 m resolution)[33], which we converted to 10 arcmin using a nearest neighbour approach. The resulting raster was overlain with the polygon map of continents, rasterized at 10 arcmin. To each pixel, we assigned the corresponding mycorrhizal type proportions, considering the prevailing combination of Bailey ecoregion × land cover in each continent. Because some ecoregions covered multiple isolated parts of continents and proportions of mycorrhizal type distribution differed in these regional areas by > 2-fold, we split these ecoregions into two or more subregions using the ArcGIS 10.2.2 Raster Calculator.

**Impact of croplands**. We estimated the effect of agriculture on mycorrhizal type distribution by substituting the proportions of mycorrhizal types in croplands (land-cover types 10, 11, 12, 20 and 30; see Supplementary Data 2) with estimates of mycorrhizal type proportions from natural land cover types in these ecoregions. For ecoregions naturally harbouring more than one vegetation type (e.g., grasslands, shrublands and forests), we considered that the expected vegetation represents a mixture of these land cover types. This resulted in an additional dataset describing combinations of vegetation as defined by ecoregion × continent × land cover without croplands (Supplementary Data 7 for continent data and Supplementary Data 8 for island data). Maps of potential mycorrhizal type distribution in a cropland-free world (Supplementary Fig. 4) were created based on this dataset, following the aforementioned procedures.

In this analysis, we did not consider forest plantations to be croplands. Therefore, changes in land cover induced by forest restoration through, for instance pine plantations or eucalypt plantations, are not addressed as vegetation changes induced by cropland cultivation. The total area occupied by AM tree plantations exceeds that of EcM tree plantations[64,65] (Supplementary Table 3), suggesting that exclusion of tree plantations leads to conservative estimates of the global reduction of EcM vegetation.

*Map validation*. The maps of the current distributions of mycorrhizal biomass fractions were validated using the datasets of forest biomass structure for Eurasia[35], global analysis of impacts on mycorrhizas on carbon vs nitrogen dynamics[19], the USA-based analysis of mycorrhizal associations conducted with remote sensing techniques[36], and the map of mycorrhizal root biomass in West Australia by Brundrett[24] (see Supplementary Fig. 2).

The data of forest biomass structure for Eurasia[35] provide information on per-plot tree species abundances for a large number of European sites. As the data contain all records obtained since the 19th century, we used only the data recorded after 1999. Using our database of plant species and associated mycorrhizal types we assigned every tree species with its mycorrhizal type (1344 data points, Supplementary Fig. 2). This provided us with a per-site data of the relative biomass of AM and EcM trees. We used these data as proxies for AM and EcM biomass fractions to compare with the data in our maps. We used the same approach for the data of Lin and co-workers[19], which represent plot-based records of vegetation structure for 100 sites across the globe accompanied with data about plant-mycorrhizal associations.

The dataset of Fisher et al.[36] provides the relative cover of AM and EcM plants from Landsat scenes centred on four sites in USA: Lilly-Dickey Woods (Indiana), long-term research site of Smithsonian Conservation Biology Institute (Virginia), Tyson Research Center Plot (Missouri), and a long-term research site of Wabikon Forest Dynamics (Wisconsin). Given that the dataset comprises forested areas only, we considered areal coverage of AM and EcM plants provides a good estimate of AM and EcM plant biomass. Using this dataset, we directly compared the AM and EcM coverage per pixel with the data of our maps.

In the datasets[19,35,36] biomass and areal fractions of AM and EcM plants are always considered to sum up to 100%. As these datasets do not provide information about non-mycorrhizal and ericoid plants, we estimated these as 5–10% and 0–20%, respectively, depending on the dominant tree association, and accordingly reduced the values of AM and EcM biomass fractions in the validation calculations. While we considered this approach to be acceptable for the validation of AM and EcM data, the data quality is not high enough to validate the NM and ErM maps.

As these three datasets[19,35,36] represent forest data, we evaluated whether all data points or raster cells[36] were indeed located in forest areas. This was done using the ESA land cover categories data[33]. All data points that were located out of the current areas registers by ESA[33] as forests were excluded from the analysis.

The West Australian map of mycorrhizal root abundance[24] provides information about the percentage of total biomass of plant roots featuring AM and EcM root colonization and about the percentage of non-mycorrhizal biomass. We used this data as a proxy for biomass fractions of AM, EcM and NM plants. In order to quantify the differences between the validation datasets and our maps, we have calculated the Mean Averaged Error (MAE) of the difference between our maps and validation datasets. MAE expresses the deviation between two spatial datasets from the 1:1 line. For AM, EcM and NM vegetation fractions MAE is 18.7%, for EcM it is 13.6%, for NM it is 4.7%, respectively. Due to a virtual absence of information about distribution of solely ErM vegetation, direct validation of the ErM maps was impossible.

To further assess the uncertainties in the maps, we examined which land use classes represent the data points that deviate from the observed data by more than 25% units. Our analysis showed that the large proportion of those deviations (60% for EcM and 40% for AM) fall into those land use classes that represent a poorly described mixture of evergreen or mixed forests and grasslands, i.e. ESA classes described as various forms of "closed to open (>15%) forest" (Supplementary Data 2). Further improvement of the ESA classification data will provide a possibility to improve precision of our maps.

Due to the rarity of datasets on field-examined mycorrhizal vegetation distributions at large special scale we had to validate our datasets used available data on plant species distribution and to accept a number of assumptions and/or recalculations in order to make the data comparable. These adjustments may have affected the quality of the validation dataset and therefore the validation.

**Map spatial uncertainty analysis**. We quantified the uncertainty in our maps of mycorrhizal vegetation fractions by applying the error propagation rules to the formulas used to calculate the biomass fractions of mycorrhizal plants per-grid cell. For this we used the data provided by refs. [35,60–63] to estimate the uncertainty associated with relative biomass of trees, shrubs, and herbaceous vegetation in each

CCI land cover class. The uncertainty in the proportion of each mycorrhizal type in a given Bailey × continent combination was set to $1/\sqrt{n}$, where $n = b + c/3 + g/20$. In this formula, $b$ is a number of literature sources describing the vegetation composition in a given Bailey ecoregion × continent combination (Supplementary Data 3), while $c$ and $g$ are the numbers of literature sources describing vegetation composition at continent level and global levels, respectively. These latter sources were given less weight, because of their lower spatial explicitness, though these sources were used only if they were providing information relevant for the combination of Bailey ecoregion × continent under consideration. This procedure was applied to the maps of the current distribution of mycorrhizal fractions (Fig. 1) and for the distribution of mycorrhizal fractions in the cropland-free world (Supplementary Fig. 5). The resulting maps of uncertainties are shown in Supplementary Figs. 4 and 6, respectively. We calculated the uncertainties in the estimations of changes in biomass fractions of mycorrhizal vegetation induced by crop cultivation and pastures (Fig. 3) by applying the error propagation rule for the mathematical operation of subtraction. The resulting uncertainty map is shown in Supplementary Fig. 7.

**Carbon stocks in the aboveground mycorrhizal biomass**. In order to estimate the amount of carbon stored globally in the biomass of plants that belong to different mycorrhizal types, we multiplied the mycorrhizal type biomass fractions by the data of the global distribution of carbon stored in aboveground plant biomass obtained from passive microwave-based satellite observations[38]. As these data have resolution of 15 arcmin, we converted our data of mycorrhizal biomass fractions to this resolution. To calculate the total amount of carbon stored in AM, EcM, ErM and NM plants (illustrated in Fig. 2, and reported per biome in the Supplementary Table 1), we summed the data on carbon stocks per unit area globally and per biome[46].

To assess uncertainty of estimations of carbon storage in mycorrhizal vegetation we used the rule of uncertainty propagation through the operation of multiplication. For this we used the uncertainty of mycorrhizal biomass fraction data and the per biome uncertainty of biomass carbon data, as provided by Liu et al.[38]. Given that the data of carbon stored in vegetation of different mycorrhizal types is a product of two datasets, each featuring a high uncertainty[38], the values of aboveground biomass carbon stored in arbuscular, ecto-, ericoid and non-mycorrhizal vegetation are locally associated with large uncertainties (e.g. Supplementary Table 1). Therefore, we recommend to use these data exclusively for large-scale estimates of biomass carbon storage. However, following the central limit theorem, the uncertainties of the total amount of carbon stored in vegetation of each mycorrhizal type are much lower: 15, 17, 1.8 and 5.5 GT for AM, EcM, ErM and NM vegetation, respectively.

**Statistical analysis**. We examined whether soil carbon content was related to the biomass fractions of AM and EcM plants, using generalized linear model regressions, with topsoil (0–20 cm) or subsoil (20–60 cm and 60–100 cm) C content per $m^2$ as a response variable, and biome and ecosystem biomass fraction of AM or EcM plants as predictors. The data for soil carbon content in the top 20 cm of soil were obtained from the ISRIC-WISE Soil Property Databases at a resolution of 30 arcsec[47]. As we were interested in relationships between AM or EcM coverage and soil carbon in natural vegetated environments, we excluded urban and agricultural areas, lakes and "Rock and Ice" areas according to the ESA land cover categories[33], from the analysis. We also excluded wetlands, inundated areas, and extremely dry regions (deserts and Mediterranean areas), because we considered that harsh abiotic conditions instead of biotic interactions are likely to shape biogeochemical cycles in these areas; and we excluded five land cover categories for which our maps showed higher uncertainties, i.e. those where the extent of forest vs grassland cover was unclear. Those land cover categories included "Tree cover, broadleaved, evergreen, closed to open (>15%)", "Tree cover, broadleaved, deciduous, closed to open (>15%)", "Tree cover, needleleaved, evergreen, closed to open (>15%)", "Tree cover, needleleaved, deciduous, closed to open (>15%)" and "Sparse vegetation (tree, shrub, herbaceous cover)".

Data for biome types were obtained from the map of terrestrial biomes[46]. To create a more balanced dataset, we combined all natural grasslands into one single category and all coniferous forests into another single category. To minimize the impacts of imprecise estimations of biome borders we excluded from forest biome areas that we recognized as grasslands according to the ESA land cover data[33]. Similarly, we excluded grassland biome areas that were fully covered by forests and shrubs according to ref. [33]. However, given that AM-EcM-NM interactions are likely to relate to C stocks in the areas featuring forest-grassland and forest-tundra mosaics, we kept such areas in the dataset. As soil carbon content data is known to feature high uncertainties[47], we opted to run the analyses at a resolution of 1 arcdegree, which allow us to grasp large-scale tendencies, while reducing the problem of P-value fallacy[66].

Our models comprised topsoil and subsoil carbon content as a response variable. As predictors, we examined the following combinations of data: (i) biome, biomass fraction of EcM plants, and their interactions, (ii) biome, biomass fraction of AM plants, and their interactions, (iii) biome, amount of carbon stored in the aboveground biomass of EcM plants (product of our EcM map and ref. [38]), and their interactions, and (iv) biome, amount of carbon stored in the aboveground biomass of AM plants (product of our AM map and ref. [38]), and their interactions.

All models were examined independently and evaluated based on the Akaike Information Criterion.

In all models, we assessed the total variance explained by the model by R-squared metric and the relative importance of each predictor using the Lindemann, Merenda and Gold (LMG) metric. This metric shows the proportion of variance explained by each of model predictors within the entire variance explained.

Given significant interactions between biome factor and biomass fractions of mycorrhizal plants (Table 1), we fitted generalized linear models with EcM or AM biomass fractions as predictors and topsoil and subsoil C stocks as response variables to the data within individual biomes. The outcome of this analysis is reported in the Supplementary Table 2. As this analysis encompasses considerable errors on both axes of this regression, we have additionally checked if a model II regression would yield qualitatively similar results and confirmed that this was the case (Supplementary Table 4).

Additionally, we examined whether the amount of carbon stored in AM and EcM plants per unit area was a better predictor of soil C stocks than the fractions of AM or EcM biomass per unit area. This analysis was performed in exactly the same manner as the analysis of the impacts of AM and EcM biomass. We assessed the resulting models using the Akaike Information Criterion and R-squared, and detected that these models were worse than the those based on the fractions of AM or EcM biomass per unit area. We also examined how fractions of AM or EcM biomass per unit area, were related to soil C-to-N ratio, and detected that these relationships had a pattern very similar to that of soil C.

**Dealing with spatial autocorrelations**. The above-described analyses were checked for spatial autocorrelation using a Moran's I metric. These tests revealed a spatial autocorrelation of residuals, which is expected given the nature of our soil and vegetation data with spatial points nested within biomes. We examined how incorporation of a spatial dependence structure affected our models. Accordingly, we ran generalized least square (GLS) models of the same sets of predictors as described in the section "Statistical analysis" of the relationship between soil carbon content and biomass distribution of mycorrhizal types, but included spatial correlation structures. Because calculation of the global spatial autocorrelation matix is computationally intensive, we chose a Monte-Carlo type approach. For each analysis we drew a random sample of 2% of the data points (1624 points) 100 times, taking care that all biomes are included into each sample, and subsequently ran a GLS accounting for autocorrelation structure of the data. For this we used the algorithms proposed by Zuur et al.[67]. The averaged outcomes of these analyses for the relationship between topsoil C, AM, and EcM fractions of vegetation biomass are shown in the Supplementary Table 5. The analyses accounting for auto-correlations yielded the same conclusions as the analyses that did not account for autocorrelations: there is a positive relationship between biomass fraction of EcM plants in vegetation and soil C, and this relationship is biome-independent (non-significant Biome × EcM interaction); in contrast, the relationship between biomass fraction of AM plants in vegetation and soil C was idiosyncratic and biome-dependent.

Including spatial coordinates explicitly in the model, or explicit accounting for autocorrelation matrix, can be problematic to interpret because covariation between spatial coordinates and environmental variables can obscure the interpretation of the relative importance of the predictors[68]. Given that the main goal of our analysis was to detect the global links between AM and EcM dominance and soil C, but not to predict this relationship in new areas or under future climatic scenarios we prioritized the interpretation of the models rather than their predictive power, and therfore report the outcomes of the models that do not account for autocorrelations (Table 1).

**Reporting summary**. Further information on research design is available in the Nature Research Reporting Summary linked to this article.

## Data availability

All the authors' data used to create the maps of mycorrhizal vegetation biomass, and all the authors' data underlying figures and tables in the manuscript text and supplementary material are available as Source Date files. All maps presented in this paper have been deposited into the DRYAD Digital Repository at https://doi.org/10.5061/dryad.47d7wm387. All the links to the publicly available datasets used within the process of assembly of the maps of mycorrhizal vegetation, in addition to the authors' data, are provided in the reference list of the paper.

## Code availability

All the codes and necessary data files used to the generate maps of mycorrhizal vegetation biomass are available at the GitHub repository at https://github.com/nasoudzilovskaia/Soudzilovskaia_NatureComm_MycoMaps.

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

## Acknowledgements

This study was supported by the vidi grant 016.161.318 issued to NAS by The Netherlands Organization for Scientific research and grants EcolChange, MOBERC and 1399PUT issued to L.T. by the Estonian Research Council. I.M. was supported by the H2020 projects GeoEssential (689443) and MAGIC (727698). C.T. was supported by a Lawrence Fellow award through the Lawrence Livermore National Laboratory (LLNL). This work was performed under the auspices of the US Department of Energy by LLNL under contract DE-AC52-07NA27344 and was supported by the LLNL-LDRD Program under Project No. 20-ERD-055. J.B.F. was supported by funding from DOE BER TES and NASA IDS. The authors thank Daisy Brickhill for editing the paper. Michala Phillips assisted J.B.F. in data transfer. J.B.F. contributed to this research, in part, from the Jet Propulsion Laboratory, California Institute of Technology, under a contract with the National Aeronautics and Space Administration, California Institute of Technology. Government sponsorship is acknowledged. The authors also thank Milagros Barcelo, Stijn Vaessen and Jinhong He who helped in compiling data on plant–mycorrhizal status.

## Author contributions

N.A.S. and C.T. generated the idea. N.A.S., P.M.v.B., L.T. and C.T. designed the study, and analysed the data. N.A.S. wrote the paper. N.A.S. and I.M. prepared the figures, with I.M. leading the role. L.T. provided mycorrhizal type assignments to vegetation. C.T., I.M., N.C.d.S. and M.v.Z. performed geoinformatics data processing. M.L.M., J.B.F. and M.C.B. provided data for map validation. All authors contributed to preparation of manuscript drafts.

## Competing interests

The authors declare no competing interests.
