## [Peer Review File · Nature Communications]

Reviewers' comments:

Reviewer #1 (Remarks to the Author):

Review of Soudzilovskaia et al. for Nature Communications, 2018

Soudzilovskaia and colleagues collate thousands of observations of plant community composition observations and the largest database of plant mycorrhizal associations ever assembled to map, for the first time, the distribution of plant mycorrhizal types on Earth. Beyond this, they relate plant mycorrhizal types to clear patterns in soil C stabilization, and quantify the potential loss of different classes of mycorrhizal associations to agricultural intensification at a global scale. This primary data search and aggregation is more than substantial. For these reasons and more, this is a one of a kind, ground-breaking analysis that represents a significant scientific breakthrough in the field.

I am largely supportive of this manuscript. I think the arguments are well made and well supported by the data. I believe their conclusions are sound. However, my support is not without reservation, and the reservation is largely methodological. I believe authors can address them, but until I know more I am apprehensive to "believe" the result of this spatial scaling.

Scaling of plant composition:

Authors aggregated data based on unique combinations of Bailey ecoregion * continent * land cover type. If these regions were uniquely plotted on a map, how fine of a scale would they be? Generally spatial aggregations are done at a fixed spatial resolution. While I find the approach here appealing, it is challenging to understand the true spatial scale of the aggregation. It would be helpful if authors could provide an estimate of the size of the average spatial aggregation unit, as well as a standard deviation.

How were the data aggregated within a cell? Authors used >1,500 unique data sources. These individual studies measure plant abundance in a multitude of different ways and at different spatial scales. Was a simple average of all compositional abundances taken per grid cell? If so, how did authors deal with non-normality of residuals when analyzing compositional data? Did you weight by observation area? by sampling effort? These methodological details are unclear, at least from the description provided on 353-372. Without a more thorough description, or a link to code, this is difficult to assess.

Validation of mycorrhizal composition maps:

While I worry about the details regarding the aggregation, an out of sample validation is a useful way to assess the accuracy of the aggregation methods without such detail. Authors show that the errors are homogeneously distributed around zero in Supplementary Figure 3, and report that most observations are within 25% of predicted values. This is important. However, equally important is the accuracy of the prediction. Can you explain 25% of the variation in out of sample data composition? 75%? R2 values would be a useful diagnostic here, and intuitive to many.

Uncertainty:

The prediction of vegetation data is not perfect. Authors also rely on multiple conversion factors to get to their final answers at scale. These conversion factors are also very uncertain. Given this, I think it is important to propagate uncertainty through this analysis using an ensemble approach. This would allow authors to report confidence intervals around their estimates. This should be done at least for the relationship between predicted and observed vegetation mycorrhizal composition and conversion factors.

Relationships between mycorrhizal type and soil carbon:

The relationships between mycorrhizal type and soil carbon (and above ground biomass for that matter) are based on spatial products, which themselves have major uncertainties. Correlations

between the ISRIC WISE predicted soil carbon and the data that are used to generate that map are never greater than 50%. Again, this is another huge source of uncertainty in these relationships not reflected in this analysis.

Finally, while authors have done an amazing job generating the most thorough mycorrhizal association list ever compiled, it is not available. Sources used are present in table 1, but there is clearly a database behind this that links mycorrhizal types to plant species identities. It is important that authors make this data publicly available, and free to use no strings attached. This resource would allow many more breakthroughs in global mycorrhizal research, global ecology and carbon cycle science.

Reviewer #2 (Remarks to the Author):

This study is a brave attempt to estimate proportions of mycorrhizal types in vegetation globally, and relate that to global carbon storage in biomass and soils. These estimates are displayed as novel maps at ~18km resolution, to fill the data gap the authors identify ('we still lack global information on the distribution of mycorrhizal types'). Guestimates of equivalent values without humans are also derived and used to conclude that humans have reduced global terrestrial carbon stocks.

If the maps and analyses are valid, this research effort would represent a significant step forward, which would be of interest to others in the field. I suspect that the estimates in the maps are probably in the right ball park, but the errors and uncertainties should be determined and presented prominently. I do not think the estimates are likely to be accurate enough to warrant 'detailed and systematic analyses of mycorrhizal biogeography and the environmental drivers thereof', as claimed by the authors (L283). The results of the analyses relating carbon stocks to mycorrhizal fractions may also be qualitatively OK, but lack of error propagation in these analyses seems problematic.

Given the uncertainties and concerns, I consider the conclusions and claims of the paper to be over-stated, to some extent.

Overall, I think this work is potentially valuable, but needs tightening up a lot, particularly in terms of stating and propagating errors, and clearly stating assumptions and limitations. I now explain in more detail.

Quantification and statement of errors and uncertainties are needed:

For example, the Abstract and L131 present estimates of carbon storage in vegetation as numbers in GT carbon, but with NO ERRORS. To compound this issue, it is illustrated in Fig.2 which is basically just a pie chart with no attempt at showing errors. I did not find estimation of these errors anywhere in the manuscript. This seems unacceptable; the errors are undoubtedly large.

Note: I am not an expert on mycorrhizas, but I suspect that a lot of simplifying assumptions are introduced by the processes described (though only vaguely) in L353-363, thus introducing considerable error into the estimates of mycorrhizal proportions.

Further crude simplifying assumptions are made in other parts of the methods, such as in determining the natural vegetation cover, in calculating expected fractions (NB not 'factions', L387!).

Multiplying by NDVI to produce biomass data is a particularly dubious procedure (L468) because NDVI is not even close to measuring biomass – it is a moderate correlate of productivity but at least two steps removed from biomass.

On the other hand, considerable effort was put into validating the data (good). L117 says 'validation revealed that the vast majority of the data (87% of the AM data points and 89% of the EcM data points) deviate by <25% from the measurements'.

On the face of it, this is not too bad, but Supp. Fig.3 shows a reasonably substantial number of deviations at or near 100%, which is a concern.

However, the validation may be affected by some of the same assumptions in both sets of estimates being compared (ie the data being validated and the data used to do the validation). For example, in any case where NDVI is used somewhere in the calculation of both estimates, the results obtained from validation will be overly optimistic because there is circularity in the validation. This issue should be fully considered and exposed, and resulting caveats made clear.

I am nervous about claims for providing 'new insights' (eg L264), given all the estimation and indirect measurement involved, plus the largish errors and lack of error propagation.

I am even more nervous about statements like 'Our mycorrhizal distribution maps provide an essential basis for detailed and systematic analyses of mycorrhizal biogeography and the environmental drivers thereof.' (L283).

It is good to try to get a reasonable global overview, as here, but the limitations must be properly recognized and uncertainty presented. Anyone considering using these data for detailed analysis etc should be made aware of the limitations.

I have concerns about the analyses:

The estimates of % mycorrhizal type are related to the estimates of aboveground C (etc; L188) without including the errors in those estimates. There is no attempt at error propagation, so these analyses seem more powerful and definitive than they should.

There is also likely to be strong spatial autocorrelation, causing non-independence of residuals. Both AIC and P-values will be strongly affected by this issue and so I have little confidence in either. This needs to be addressed.

In the expected fractions before human influence, should I be concerned that assumptions are made about eg depth distribution of mycorrhizas in the soil, when we only have data from a time when enhanced global N deposition (among other anthropogenic phenomena) has been occurring for decades, potentially altering these patterns?

Concern about causality, and therefore the validity of the stated implications relating to human modification of landscapes:

Are the mycorrhizas causing or responding? If responding, then is it to C content of the soil, or to other influences such as N availability? I asked these questions on reading the abstract, and was getting increasingly annoyed by the lack of mention of this issue, until finally in L253 this point is addressed: 'Although it can be argued that high abundance of EcM plants is a consequence rather than a driver of soil C accumulation, a large body of recent findings provides evidence that EcM symbionts may be the key drivers of topsoil carbon accumulation through two interacting mechanisms...' I suggest this point should be made and justified in the introduction. And even so, it is not convincing about whether both soil C and mycorrhizas are responding to another change, such as N deposition (stimulating eg bacterial activity).

This point is relevant to conclusions such as 'We show that human-induced transformations of Earth's ecosystems reduced ectomycorrhizal vegetation, with potential knock-on effects on terrestrial carbon stocks.' (L36).

Specific or minor comments

Language consistency. Eg for the focal associations, the plural is sometimes 'mycorrhiza' and sometimes 'mycorrhizas'.

Also, mostly the word 'data' is used as plural (correct) but quite often it is used as singular. These and other minor language issues need sorting out.

The maps are too small in many cases to see much of the pattern, though zooming to 500% does help in eg Fig.4.

L176 'positive impact of greater abundance of EcM relative to AM trees on soil carbon content' is not clear. Positive could be a value judgment (in which case it is not clear) or could mean an increase in soil carbon content (in which case say so more clearly).

From L 129 it looks like Fig.3 shows data from Liu et al 2015, rather than being original to this study. If so, this should be stated clearly in the figure legend. If not then proper explanation needs to be given.

Are 'generalized linear model (glm) regressions of the Gaussian family' not just ordinary OLS regressions? So why is it necessary to use Cragg and Uhler's pseudo R-squared metric instead of standard R²? And why estimate relative importance using the Lindemann, Merenda and Gold (LMG) metric rather than just partial R²?

In Table 1, I suggest also adding either partial R² or explanation of how to interpret %LMG values. Table 1 is also missing important information such as sample size.

Also, are the results in Table 1 based on Type I (sequential adding of terms) or Type III (deletion from final model) approaches? This distinction is very important for interpreting the results.

Supp Fig.3 does not state the units on X. Guess %. If so, % of what, exactly?
More generally, explanation of some of the Figures can be improved.

Reviewer #3 (Remarks to the Author):

This work aims to connect the distribution of different types of mycorrhizal associations to plant and soil carbon storage at the global scale, with a focus on comparing ectomycorrhizal, ericoid, and arbuscular mycorrhizal plant functional groups. While I enjoyed the scope of this work, I feel that more can be done to (1) demonstrate the novelty of this work in comparison to previous efforts, (2) control for other environmental and/or plant variables that may affect plant and soil C and to (3) map uncertainties in plant and soil C projections. I also don't necessarily think that the agricultural land-use change comparison is a strong one and at least should be considered in light of other global change factors. I have specific comments below.

Novelty: The introduction of this work needs to distinguish how having maps of mycorrhizal distributions can improve on previous carbon estimates by Averill et al. 2014 in Nature, and mycorrhizal distributions in Menzel et al. 2016 in Perspectives in Plant Ecology, Evolution and Systematics, and Steidinger et al. 2018 in biorxiv. In In 64 you state that we lack global maps of mycorrhizal types but Tedersoo et al. 2014 maps ectomycorrhizal distributions and Opik, Davison and other have mapped arbuscular mycorrhizal distributions. The distinction is that these previous publications have not linked mycorrhizal types to aboveground or soil C, which needs to be made clearer here.

Other variables such as climate, soil resources, and soil texture may affect soil carbon and plant productivity. These are mainly assumed to be captured by biome types, but worldwide maps of

these variables are common. They should be incorporated into the statistical model in order to demonstrate that mycorrhizal type itself is driving patterns of plant biomass and soil C and that other variables such as climate are not driving both the mycorrhizal distributions and these ecosystem parameters. This is especially important as data presented in Table 1 seems to indicate that mycorrhizal type is ancillary to biome in predicting soil C storage in all cases.

Uncertainty. From the map in Supplemental Figure 2, it seems like many areas of the western hemisphere, central Asia, and Africa are poorly validated in this dataset. This is not surprising because these are often the areas with lower sampling effort. However, this may create uncertainty/error in your models. Therefore, another map of this uncertainty should be generated for all of the ecosystem C estimates that you present here. While this may show that we don't know much about the constraints of C in these areas, it is also an opportunity to point out future research sampling efforts. This should be able to replace the current Figure 2, which is already covered in the text.

Lns 322-324 suggest that AM fungi are not functional in agricultural systems. This is inaccurate as many studies have surveyed the richness, composition, and functioning of AM fungi in these systems (e.g., Verbruggen et al. 2015 *Soil Biology and Biochemistry*). At the very least other global change drivers (e.g., N deposition, climate change, etc.) should be compared to land-use change effects on these distributions. I suspect that an accurate accounting of AM fungi in agricultural systems may lessen many of these findings or fall within the range of uncertainty for the other global maps.

Lns 414 – R scripts should be available in the manuscript as a supplement.

Dear Editors, Dear Reviewers,

We were very happy to receive a positive evaluation of our manuscript “Global mycorrhizal plant distribution linked to terrestrial carbon stocks”. We would like to thank all three reviewers for insightful and constructive suggestions for improvement of the manuscript, and to thank the Editor for the resubmission deadline extension granted by email (please see our communication with Dr Andrew on 25-2-2019). Below we outline the changes that we have introduced into the paper following the reviewer’s comments point by point. In this letter we indicate which lines in the manuscript have been changed. In the manuscript itself we have highlighted the changed lines with blue colour. As requested by the Editor and reviewers we provide codes used to assemble the maps of mycorrhizal vegetation, and the data underlying the manuscript Figures and Tables. All the data is available at a public GitHub repository https://github.com/nasoudzilovskaia/Soudzilovskaia_NatureComm_MycoMaps .

With best regards,
Nadejda Soudzilovskaia and co-authors

Reviewer #1 (Remarks to the Author):

R1.0. Soudzilovskaia and colleagues collate thousands of observations of plant community composition observations and the largest database of plant mycorrhizal associations ever assembled to map, for the first time, the distribution of plant mycorrhizal types on Earth. Beyond this, they relate plant mycorrhizal types to clear patterns in soil C stabilization, and quantify the potential loss of different classes of mycorrhizal associations to agricultural intensification at a global scale. This primary data search and aggregation is more than substantial. For these reasons and more, this is a one of a kind, ground-breaking analysis that represents a significant scientific breakthrough in the field.

Re: Thank you very much for your support!

R1.1. I am largely supportive of this manuscript. I think the arguments are well made and well supported by the data. I believe their conclusions are sound. However, my support is not without reservation, and the reservation is largely methodological. I believe authors can address them, but until I know more I am apprehensive to "believe" the result of this spatial scaling.

Scaling of plant composition:

*Authors aggregated data based on unique combinations of Bailey ecoregion * continent * land cover type. If these regions were uniquely plotted on a map, how fine of a scale would they be? Generally spatial aggregations are done at a fixed spatial resolution. While I find the approach here appealing, it is challenging to understand the true spatial scale of the aggregation. It would be helpful if authors could provide an estimate of the size of the average spatial aggregation unit, as well as a standard deviation.*

Re: We thank the reviewer for the inquiry. We were in fact not clear enough in our description of how we combined the various datasets, each having different spatial resolution. We have clarified this now in the lines 479-492, and in we provide the annotated code used to assemble the maps, in a public GitHub repository https://github.com/nasoudzilovskaia/Soudzilovskaia_NatureComm_MycoMaps

We have converted the data of AM, EcM, ErM, NM plant biomass per combination of Bailey ecoregion * Land cover type * Continent, presented in the Supplementary Tables 5-8 into raster maps following the steps outlined below:

- (1) We overlapped the raster map of Bailey ecoregions (10 arcmin) with the raster ESA CCI land cover dataset 30. As the latter dataset originally had a resolution of 300m, we aggregated it to 10 arcmin using a nearest neighbour approach.
- (2) The resulting raster was overlapped with the polygon map of continents, rasterized at 10 arcmin.
- (3) In the resulting map (spatial resolution 10 arcmin), we assigned data to every pixel according to the combination of Bailey ecoregion *Land cover *Continent as prescribed by the outcome of our literature analysis (Supplementary Tables 5-8).

We report our minimal aggregation unit to be 10 arcmin. Providing the standard deviation resulting from spatial aggregation is unfortunately not possible in this case, as the only map that has been truly aggregated is the ESA land cover map, which is categorical.

R1.2. How were the data aggregated within a cell? Authors used >1,500 unique data sources. These individual studies measure plant abundance in a multitude of different ways and at different spatial scales. Was a simple average of all compositional abundances taken per grid cell? If so, how did authors deal with non-normality of residuals when analyzing compositional data? Did you weight by observation area? by sampling effort? These methodological details are unclear, at least from the description provided on 353-372. Without a more thorough description, or a link to code, this is difficult to assess.

RE: For each category, we used relative abundances of plant species averaged across different data points, with equal weight given to all. Given the plethora of original methods, site size and plant density, a weighting approach would introduce more biases and complexity, and was thus not used. We have clarified this point in lines 423-426. The R scripts used to assemble the maps are available at https://github.com/nasoudzilovskaia/Soudzilovskaia_NatureComm_MycoMaps.

R1.3. Validation of mycorrhizal composition maps:

While I worry about the details regarding the aggregation, an out of sample validation is a useful way to assess the accuracy of the aggregation methods without such detail. Authors show that the errors are homogeneously distributed around zero in Supplementary Figure 3, and report that most observations are within 25% of predicted values. This is important. However, equally important is the accuracy of the prediction. Can you explain 25% of the variation in out of sample data composition? 75%? R2 values would be a useful diagnostic here, and intuitive to many.

Re: We agree with the reviewer's comment about the importance of reporting the accuracy of the maps aggregation. We have calculated the difference between the Root Mean Square Deviations of the data presented in our maps and in validation datasets, and reported them in the lines 150-152 and 543-559 in the manuscript text, in lines 18-20 in the Supplementary Data file, and in the Supplementary Data Figure 3].

We cannot entirely agree that R^2 is the best method to assess the similarity between two datasets. Please consider an example of a comparison of two datasets: A: [2,3,1,4,5,1] and B:[200, 300, 100, 400, 500, 100]. The dataset A perfectly predicts the variation in the dataset B, with $R^2=1$. However, the datasets A and B are different, with values in B ten orders of magnitude larger than in A. Instead, we calculated the Mean Absolute Error and Root Mean Square Deviation (RMSD), two more commonly used methods to compare spatial datasets (Refs: Pontius J. et al. (2008) "Components of information for multiple resolution comparison between maps that share a real variable". *Environmental and Ecological Statistics*. 15: 111–142.; Hyndman, R. and Koehler A. (2005). "Another look at measures of forecast accuracy"). We used both approaches, and found out that in our case they provide similar outcomes. Following discussions outlined in the work of Hyndman, R. and Koehler A. (see the full reference above) we opted to report the RMSD values in the manuscript because it is more

intuitively understandable, providing a direct estimation of the standard deviation of the data. We have added these values to Extended Data Figure 3.

R1.4. Uncertainty:

The prediction of vegetation data is not perfect. Authors also rely on multiple conversion factors to get to their final answers at scale. These conversion factors are also very uncertain. Given this, I think it is important to propagate uncertainty through this analysis using an ensemble approach. This would allow authors to report confidence intervals around their estimates. This should be done at least for the relationship between predicted and observed vegetation mycorrhizal composition and conversion factors.

Re: Thank you for raising this point. We agree that it is important to provide an estimation of uncertainties of the maps. Following the reviewer's suggestion, we have used the relationships between the predicted and observed mycorrhizal vegetation to estimate uncertainties, see the reply to R1.3 for details. We have added a discussion about the maps uncertainties and their implications to the manuscript in the lines 136-144, and 555-559 and we provide a total estimation of the data uncertainty as the averaged RMSD values for each of the AM, EcM, ErM and NM vegetation maps in the lines 140-141, 550-551, and in the Caption of the Figure 1]. Furthermore, we use the RMSD values of the differences between the maps and validation datasets to estimate the errors in the calculations of carbon stored in vegetation (the point also raised by the reviewer 2, see below under R2.1)

R1.5. Relationships between mycorrhizal type and soil carbon:

The relationships between mycorrhizal type and soil carbon (and above ground biomass for that matter) are based on spatial products, which themselves have major uncertainties. Correlations between the ISRIC WISE predicted soil carbon and the data that are used to generate that map are never greater than 50%. Again, this is another huge source of uncertainty in these relationships not reflected in this analysis.

Re: We agree that data from ISRIC have their own uncertainties, and these uncertainties add ambiguity to the analysis of the relationship between mycorrhizal dominance and soil C. We have added a discussion of this point into the manuscript (lines 244-254). The high level of these uncertainties is also a reason why we analyse the data at the resolution of 1 arcdegree. As soil carbon content data is known to feature high uncertainties we opted to run the analyses at a coarse resolution, which allows grasping large scale tendencies, while reducing the problem of P-value fallacy, as discussed in the lines 620-622 of the current manuscript version (lines 516-518 in the previous version).

We consider that these uncertainties equally apply to the analysis of AM vs soil C as well as to that of the EcM vs soil C, due to the fact that the analysis is based on the same geographical points. Therefore, given the clarified research question underlying this analysis being "Do AM- and EcM-dominant vegetation differently relate to soil C?" (see under R3.2, for the discussion on the need of such clarification), we consider that these uncertainties did not affect our conclusions originating from the analysis.

1.6. Finally, while authors have done an amazing job generating the most thorough mycorrhizal association list ever compiled, it is not available. Sources used are present in table 1, but there is clearly a database behind this that links mycorrhizal types to plant species identities. It is important that authors make this data publicly available, and free to use no strings attached. This resource would allow many more breakthroughs in global mycorrhizal research, global ecology and carbon cycle science.

Re: Thank you for the high valuation of the quality of our dataset on mycorrhizal associations of vascular plants. This dataset will be a subject of a separate publication, currently prepared for an invited submission to *New Phytologist* (the anticipated submission date is mid-April). The dataset also includes data about levels of plant root colonisation by mycorrhizal fungi

and expert judgements on data reliability, much of which are contributed by several additional researchers not involved in the current paper. Therefore we would prefer to keep these two publications separate.

Reviewer #2 (Remarks to the Author):

R2.0. This study is a brave attempt to estimate proportions of mycorrhizal types in vegetation globally, and relate that to global carbon storage in biomass and soils. These estimates are displayed as novel maps at ~18km resolution, to fill the data gap the authors identify ('we still lack global information on the distribution of mycorrhizal types'). Guestimates of equivalent values without humans are also derived and used to conclude that humans have reduced global terrestrial carbon stocks.

If the maps and analyses are valid, this research effort would represent a significant step forward, which would be of interest to others in the field. I suspect that the estimates in the maps are probably in the right ball park, but the errors and uncertainties should be determined and presented prominently. I do not think the estimates are likely to be accurate enough to warrant 'detailed and systematic analyses of mycorrhizal biogeography and the environmental drivers thereof', as claimed by the authors (L283). The results of the analyses relating carbon stocks to mycorrhizal fractions may also be qualitatively OK, but lack of error propagation in these analyses seems problematic. Given the uncertainties and concerns, I consider the conclusions and claims of the paper to be overstated, to some extent. Overall, I think this work is potentially valuable, but needs tightening up a lot, particularly in terms of stating and propagating errors, and clearly stating assumptions and limitations. I now explain in more detail.

Re: Thank you very much for the recognition of the potential value of our work as well as for the detailed suggestions to improve the manuscript. We have addressed these comments in the most comprehensive way.

R1.1. Quantification and statement of errors and uncertainties are needed:

For example, the Abstract and L131 present estimates of carbon storage in vegetation as numbers in GT carbon, but with NO ERRORS. To compound this issue, it is illustrated in Fig.2 which is basically just a pie chart with no attempt at showing errors. I did not find estimation of these errors anywhere in the manuscript. This seems unacceptable; the errors are undoubtedly large.

RE: We agree with this point. We have calculated errors associated with the estimations of carbon storage in mycorrhizal vegetation and added this information to the manuscript [lines 34-35 in the Abstract and 157-158 in the main text]. Hereto we used the following approach:

(1) For each type of mycorrhizal vegetation we have calculated the mean uncertainty of the maps using the validation datasets. For each individual dataset used for maps validation procedure we estimated an average deviation between the mycorrhizal map data and the data in the validation dataset, by calculating a mean value of Root Mean Square Deviation (RMSD) at 90% confidence interval. This value provides an estimation of the standard deviation of the maps.

(2) As the values of carbon stored in mycorrhizal vegetation have been calculated for each grid cell as a product of a mycorrhizal biomass fraction and the satellite-derived estimate of carbon stored in vegetation, we propagated uncertainty using the standard deviation following the approach of Goodman (1960) (Reference: Goodman, L. (1960). "On the Exact Variance of Products". *Journal of the American Statistical Association*. 55 (292): 708–713):

$$\sigma = \sqrt{\left(\frac{RMSD(M)}{Mean(M)}\right)^2 + \left(\frac{\sigma(C)}{Mean(C)}\right)^2}$$

where:

$RMSD(M)$ - is an estimation of standard deviation of mycorrhizal biomass fraction values, taken as validation RMSD values at 90% confidence interval;
 $Mean(M)$ – is a mean value of mycorrhizal biomass fractions, calculated based on the maps of individual mycorrhizal types (Figure 1);
 $\sigma(C)$ - is a standard deviation of the data of carbon stored in Earth vegetation biomass taken from the data of Liu et al. (2015);
 $Mean(C)$ - is a mean value of the data of carbon stored in Earth vegetation biomass taken from the data of Liu et al. (2015).

We added the description of the calculation procedure in the lines 571-590. Concerning the Figure 2, we agree that it does not add much information to the manuscript. Given that showing errors of percentages is ambiguous in this case (because they would relate to both uncertainties of our estimations of fraction of carbon stored within mycorrhizal vegetation, as well as to the total amount of carbon stored in the Earth data, i.e. the data of Liu et al, 2015) we opted to remove this figure from the manuscript.

R2.2. Note: I am not an expert on mycorrhizas, but I suspect that a lot of simplifying assumptions are introduced by the processes described (though only vaguely) in L353-363, thus introducing considerable error into the estimates of mycorrhizal proportions. Further crude simplifying assumptions are made in other parts of the methods, such as in determining the natural vegetation cover, in calculating expected fractions (NB not 'factions', L387!).

Re: We recognize that our work encompasses simplifications. However, we would like to stress that as the reviewer suggests, the creation of relatively coarse resolution maps always requires the application of simplifications. We believe that in our case the simplifications are not overly crude (as suggested by the validation outcomes), because we essentially combine information about plant species distribution to derive distribution of mycorrhizal types, which are in the majority of cases strongly linked to the plant growth forms (*ref: Tedersoo, L. & Brundrett, M. C. in Biogeography of mycorrhizal symbiosis Ecological studies, Springer, 2017*). For the distribution of those a high resolution data has been used (the maps data, representing the natural vegetation cover at resolution of 300m). We consider the largest source of uncertainty in our calculations to be the conversion of the growth form information into biomass fractions. This conversion is done based on multiple data sources, but still represents an important simplification of reality. However, this is the only alternative to using NDVI, which is (as reviewer discusses in the next comment) a much worse option. We agree with the reviewer's suggestion to discuss how these simplifications affect maps quality. We added the discussion in 136-141.
We have also corrected "factions" for "fractions".

R2.3. Multiplying by NDVI to produce biomass data is a particularly dubious procedure (L468) because NDVI is not even close to measuring biomass – it is a moderate correlate of productivity but at least two steps removed from biomass.

Re: We entirely agree with this point, and we do not use NDVI to calculate biomass, exactly for these reasons. The single mentioning of NDVI in the manuscript is an error, remaining from an early draft. We apologize for the confusion caused by this. We have removed this line as irrelevant.

R2.4. On the other hand, considerable effort was put into validating the data (good). L117 says 'validation revealed that the vast majority of the data (87% of the AM data points and 89% of the EcM data points) deviate by <25% from the measurements'. On the face of it, this is not too bad, but Supp. Fig.3 shows a reasonably substantial number of deviations at or near 100%, which is a concern.

Re: Thank you for the positive evaluation of our efforts to validate the data. Concerning the Supplementary Figure 3, the only validation showing a few points (out of several thousands) deviating by close to 100% is that based on the data by Brundrett et al. We have closely examined these datapoints and detected that these were caused by a small bug in the script processing the Australian maps. Initially that script incorrectly accounted for discontinued vegetation cover. Correction of this bug has solved this problem (see new version of the validation histograms submitted in this round). We have carefully checked whether the corrected procedure of the Australian maps processing affected the other analyses and conclusions, and detected no impacts. All data submitted in this round are based on the corrected scripts.

R2.5. However, the validation may be affected by some of the same assumptions in both sets of estimates being compared (ie the data being validated and the data used to do the validation). For example, in any case where NDVI is used somewhere in the calculation of both estimates, the results obtained from validation will be overly optimistic because there is circularity in the validation. This issue should be fully considered and exposed, and resulting caveats made clear.

Re: We agree that our validation datasets are not perfect either, and might introduce errors in validation. We added a discussion on this into the manuscript (lines 555-559). The NDVI has not been used in any parts of our analysis, see the reply to the point R2.3.

R2.6. I am nervous about claims for providing ‘new insights’ (eg L264), given all the estimation and indirect measurement involved, plus the largish errors and lack of error propagation.

I am even more nervous about statements like ‘Our mycorrhizal distribution maps provide an essential basis for detailed and systematic analyses of mycorrhizal biogeography and the environmental drivers thereof.’ (L283).

It is good to try to get a reasonable global overview, as here, but the limitations must be properly recognized and uncertainty presented. Anyone considering using these data for detailed analysis etc should be made aware of the limitations.

Re: We have added a quantification on of the map uncertainties to the manuscript text (lines 140-141, Caption Figure 1) and we added a discussion on the sources of these uncertainties and their implications for future maps use (lines 136-144 in the main manuscript text and lines 555-559 in the methods). Concerning the new statement of the novel insights (previously line 264) , we have re-phrased it as “new qualitative insights” (line 305 in the new version of the manuscript).

Concerning the statement “Our mycorrhizal distribution maps provide an essential basis for detailed and systematic analyses...”, we have removed the word “detailed” , rephrased the sentence acknowledging the uncertainties (lines 324-326), and explained why our maps provide important data for such analyses (lines 326-329). Though not perfect, our maps are the first global maps of mycorrhizal vegetation based on factual data, and not on modelling of a vegetation-to-environment relationships . Therefore, our maps provide independent data for examining drivers of mycorrhizal distribution and their impacts on ecosystem functioning, without introducing a circular reasoning caused by the use of environmental data.

R2.7. I have concerns about the analyses:

The estimates of % mycorrhizal type are related to the estimates of aboveground C (etc; L188) without including the errors in those estimates. There is no attempt at error propagation, so these analyses seem more powerful and definitive than they should.

Re: There seem to be some misunderstanding here. In the analysis of a relationship between soil C and fractions of AM or EcM plant biomass in vegetation (previously lines 187-201, as referred by the reviewer; in the new version – lines 209-211) we did not use the estimates of

the aboveground C. These analyses are directly based on the data of mycorrhizal maps, as presented in the Figure 1.

However we agree with the general message of this comment, concerning the necessity to include a discussion about the errors associated with the maps and their implications. Please see lines 244-254 and additionally lines 136-144.

R2.8. There is also likely to be strong spatial autocorrelation, causing non-independence of residuals. Both AIC and P-values will be strongly affected by this issue and so I have little confidence in either. This needs to be addressed.

Re: Thank you for raising this point. We have added the information about autocorrelation analysis in the manuscript (see lines 651-678, which constitute a new sub-chapter in the Methods section “Dealing with autocorrelations”, and new Table 11 in the Supplementary Data, lines 54-61 in the Supplementary Data file). We have re-run our analyses accounting for autocorrelations using the *gls* function from the *nlme* R package. This re-analysis indicated that all previously detected relationships hold, and even strengthen, due to the removal of the residual noise caused by autocorrelations. Thus the presence of autocorrelation did not affect our conclusions. However, the main goal of our analysis is to detect the global links between AM and EcM dominance and soil C, but not to predict this relationship in new areas or under future climatic scenarios. Including spatial coordinates explicitly in the model, or explicit accounting for autocorrelation matrix, can be problematic to interpretation because covariation of spatial coordinates with environmental variables can obscure the interpretation of the relative importance of the predictors (REFs: Dormann CF (2007) *Effects of incorporating spatial autocorrelation into the analysis of species distribution data. Global Ecology and Biogeography*, 16, 129-138; Miller J, Franklin J, Aspinall R (2007) *Incorporating spatial dependence in predictive vegetation models. Ecological Modelling*, 202, 225-242). Further, we also rely on the conclusions of Kunn and Dorman (2012) (“Less than eight (and a half) misconceptions of spatial analysis”, *J. of Biogeography*, <https://doi.org/10.1111/j.1365-2699.2012.02707.x>), who argue that spatial autocorrelation should be explicitly accounted for in studies of spatial ecological process, such as dispersal, migration, territorial behaviour etc, but are less relevant in studies like ours, where a non-spatial relationship between a variable and environmental conditions is being examined. Taking into account all these considerations, we prioritized the interpretation of the models rather than their predictive power, and opted to report in the main text the outcomes of the model that do not account for autocorrelations and to show the information about impacts of autocorrelations in the Supplementary Data (Supplementary Table 11).

R.2.9. In the expected fractions before human influence, should I be concerned that assumptions are made about eg depth distribution of mycorrhizas in the soil, when we only have data from a time when enhanced global N deposition (among other anthropogenic phenomena) has been occurring for decades, potentially altering these patterns?

Re: We agree that nitrogen deposition is another important factor that have caused shifts in distributions of mycorrhizal vegetation. These dynamics are currently object of a very active line of research (e.g. Averill et al. 2018 “Continental-scale nitrogen pollution is shifting forest mycorrhizal associations and soil carbon stocks”, at *Global Change Biology*, and a number of other papers), and are not addressed in our paper. We have clarified this in the lines 279-285. Given that nitrogen deposition is likely to reduce ectomycorrhizal vegetation and enhance non-mycorrhizal or arbuscular mycorrhizal vegetation, this human-induced stressor changes mycorrhizal distributions in the same direction as agricultural practices – replacing

ectomycorrhizal plants by arbuscular mycorrhizal plants. In this respect by limiting our analysis to agricultural practices we underestimate the anthropogenic influence on mycorrhizal re-distributions.

R2.10. Concern about causality, and therefore the validity of the stated implications relating to human modification of landscapes: Are the mycorrhizas causing or responding? If responding, then is it to C content of the soil, or to other influences such as N availability? I asked these questions on reading the abstract, and was getting increasingly annoyed by the lack of mention of this issue, until finally in L253 this point is addressed: 'Although it can be argued that high abundance of EcM plants is a consequence rather than a driver of soil C accumulation, a large body of recent findings provides evidence that EcM symbionts may be the key drivers of topsoil carbon accumulation through two interacting mechanisms...' I suggest this point should be made and justified in the introduction. And even so, it is not convincing about whether both soil C and mycorrhizas are responding to another change, such as N deposition (stimulating eg bacterial activity).

Re: Thank you for this comment. We have moved the discussion on causality into the introduction lines 62-69. We were discussing alternative drivers of soil C and mycorrhizal vegetation, including nitrogen availability in the lines 292-294. We added a discussion on the potential impacts of nitrogen deposition into the lines potential impacts of N deposition on bacterial activity into the discussion (lines 281-285).

R2.11. This point is relevant to conclusions such as 'We show that human-induced transformations of Earth's ecosystems reduced ectomycorrhizal vegetation, with potential knock-on effects on terrestrial carbon stocks.' (L36).

Re: Concerning this comment, please see also our response to the R2.9, and R2.10. We agree with the point of uncertainty concerning the impacts on soil C, and discuss this in the lines 291-295, 298-301 added to the manuscript. However, human-induced nitrogen deposition is likely to replace ectomycorrhizal vegetation by arbuscular mycorrhizal vegetation, so it acts in the same direction as agricultural practices. Given that our work shows a general trend of higher carbon storage in plant biomass in arbuscular mycorrhiza-dominated ecosystems and higher carbon storage in soil (but lower in plant biomass) in ectomycorrhizal dominated ecosystems, we consider that the conclusion about the human-induced transformations of ecosystems affecting terrestrial carbon stocks via mycorrhiza – associated pathway is still valid.

R2.12. Specific or minor comments

Language consistency. Eg for the focal associations, the plural is sometimes 'mycorrhiza' and sometimes 'mycorrhizas'.

Also, mostly the word 'data' is used as plural (correct) but quite often it is used as singular.

These and other minor language issues need sorting out.

Re: Thanks, we have corrected these issues.

R2.13. The maps are too small in many cases to see much of the pattern, though zooming to 500% does help in eg Fig.4.

Re: The maps shown in the manuscript figures have purely illustrative purposes and have the size allowed by the journal page. After the manuscript acceptance, we will make the original raster files available to the scientific community.

R2.14. L176 'positive impact of greater abundance of EcM relative to AM trees on soil carbon content' is not clear. Positive could be a value judgment (in which case it is not clear) or could mean an increase in soil carbon content (in which case say so more clearly).

Re: We have rephrased this statement (lines 198-200).

R2.15. From L 129 it looks like Fig.3 shows data from Liu et al 2015, rather than being original to this study. If so, this should be stated clearly in the figure legend. If not, then proper explanation needs to be given.

Re: We have rephrased this statement (lines 155-158). Please note that in the new version of the manuscript the figure that the reviewer refers to is Figure 2.

R2.16. Are 'generalized linear model (glm) regressions of the Gaussian family' not just ordinary OLS regressions? So why is it necessary to use Cragg and Uhler's pseudo R-squared metric instead of standard R²? And why estimate relative importance using the Lindemann, Merenda and Gold (LMG) metric rather than just partial R²? In Table 1, I suggest also adding either partial R² or explanation of how to interpret %LMG values.

Re: We agree with the reviewer that a generalized linear model of a Gaussian family is the same as an OLS regression. Therefore Cragg and Uhler's pseudo R-squared metric for Gaussian GLM is exactly the same as R². We have re-run the analysis using the OLS, and changed the descriptions throughout the text and in the captions of the Table 1 and Supplementary Table 10.

Considering the use of LMG metric, we have chosen it because it is an intuitively understandable metric, which shows the proportion of variance explained by each of model predictors. It is also independent of the sequence in which the predictors appear in the model, which is the main caveat of partial R² estimations. LMG metric is applicable to GLM as well as to OLS, and in case of Gaussian GLM with an 'identity' link (the case of our analysis in the initially submitted manuscript), it yields exactly the same result. We have added an explanation about how to interpret the LMG in the lines 221-222, and 634-635.

R2.16. Table 1 is also missing important information such as sample size.

Re: Thanks, we have added this info into the caption of the Table 1.

R2.17. Also, are the results in Table 1 based on Type I (sequential adding of terms) or Type III (deletion from final model) approaches? This distinction is very important for interpreting the results.

Re: The results in Table 1 are based on Type I ANOVA. We have added this information into the caption of the Table 1.

R2.18. Supp Fig.3 does not state the units on X. Guess %. If so, % of what, exactly?

Re: Yes, indeed it is % - i.e. difference in the values of the mycorrhizal maps presented in the manuscript (expressed as %), and data in validation dataset (also expressed as %). We have added an explanation this information into the caption of the Supplementary Figure 3 (lines 13-20 in the Supplementary Data file).

R2.19. More generally, explanation of some of the Figures can be improved.

Re: We have also improved the captions of the Figure 1, and Figure 2(Figure 3 in the previous version of the manuscript).

Reviewer #3 (Remarks to the Author):

R3.0. This work aims to connect the distribution of different types of mycorrhizal associations to plant and soil carbon storage at the global scale, with a focus on comparing ectomycorrhizal, ericoid, and arbuscular mycorrhizal plant functional groups. While I enjoyed the scope of this work, I feel that more can be done to (1) demonstrate the novelty of this work in comparison to previous efforts, (2) control for other environmental and/or plant variables that may affect plant and soil C and to (3) map uncertainties in plant and soil C projections. I also don't necessarily think that the agricultural land-use change comparison is a strong one and at least should be considered in light of other global change factors. I have specific comments below.

Re: Thank you very much for the positive assessment and constructive criticism. Below we address the comments one by one.

R.3.1. Novelty: The introduction of this work needs to distinguish how having maps of mycorrhizal distributions can improve on previous carbon estimates by Averill et al. 2014 in Nature, and mycorrhizal distributions in Menzel et al. 2016 in Perspectives in Plant Ecology, Evolution and Systematics, and Steidinger et al. 2018 in biorxiv. In In 64 you state that we lack global maps of mycorrhizal types but Tedersoo et al. 2014 maps ectomycorrhizal distributions and Opik, Davison and other have mapped arbuscular mycorrhizal distributions. The distinction is that these previous publications have not linked mycorrhizal types to aboveground or soil C, which needs to be made clearer here.

Re: Thank you for this comment. We have added into the introduction a discussion of the novelty of our work in comparison to the studies of Tedersoo et al (2014) Davidson, Opik et al (2015) and Menzel et al (2016) (lines 72-76). These studies, as well as the very interesting study of Steidinger et al. (available only as a pre-print), have mapped the distributions of arbuscular and ectomycorrhizal plant or fungal species, which are very weak predictors of the biomass of mycorrhizal plants. We have clarified this in the lines 75-76. In contrast, our maps present the first attempt to map the biomass of mycorrhizal plants based on field data, providing the most direct proxy to quantify mycorrhizal impact on ecosystem functioning. We consider that leverage of such maps constitutes an important scientific advance in itself, as the maps provide multiple possibilities for analysis of ecosystem functioning in relation to mycorrhizas. We have added a discussion on this in lines 263-270 and 324-329.

Following the suggestion of the reviewer, we also clarified the advances of our work in comparison to the work of Averill et al (2014): our work is unique and novel because (1) for the first time we link the biomass distribution of mycorrhizal vegetation to both aboveground C and soil C lines 303-306; (2) because the provide the first global quantification of the relationships between the dominance of mycorrhizal vegetation an soil C across soil depths, based on the global gridded data (lines 204-205, 287-292).

R.3.2. Other variables such as climate, soil resources, and soil texture may affect soil carbon and plant productivity. These are mainly assumed to be captured by biome types, but worldwide maps of these variables are common. They should be incorporated into the statistical model in order to demonstrate that mycorrhizal type itself is driving patterns of plant biomass and soil C and that other variables such as climate are not driving both the mycorrhizal distributions and these ecosystem parameters.

Re: We thank the reviewer for this comment, We agree that our models accounting for mycorrhizal type and biome miss some more detailed information about soil resources unrelated to climate, such as soil texture, and it does not fully account for the effect of climate. In order to understand the full puzzle of soil C accumulation, and the relative importance of mycorrhiza in explaining soil C patterns (as suggested by the reviewer) we would indeed need to assess all potential drivers of soil C and their interactions, accounting for their, currently unknown, hierarchy. However, this is not exactly the question that we address in this paper. Instead we aim to quantitatively test if the dominance of different mycorrhizal types differently relates to the global patterns in soil C (see the lines 84-85 in the initially submitted version of the manuscript, and lines 95-98 in the current manuscript version). We detect that while for EcM there is a clearly positive pattern, for AM there is no pattern (lines 224-242). In our analysis, all the unaccounted variables such as soil texture, contribute to the residuals. Given that despite the large residuals, we still see a pattern, we do not entirely agree that adding additional variables to the analysis, or decomposing the integrative characteristic of “biome” into individual aspects of soil and climate would

strengthen the conclusions of our analysis, as the added complexity would increase the ambiguity of the procedure, and complicate the interpretation of the results.

While we think that the reviewer's suggestion refers to a different question than we aimed to pose in the manuscript, we agree with the point that we need to properly discuss the complex nature of the factors underpinning soil C. This discussion is provided in the lines 62-69 (moved forward in response to the request 2.10 of the Reviewer 2) , and the added lines 279-283, 298-301.

We also additionally stress that the current analysis is unable to detect the causality of the impacts of distinct factors on soil C (lines 292-295). We however also believe that in answering the question of causality of soil C patterns, within a regression-like analysis, i.e. glm or random forest techniques, is impossible. As a former analysis is unable to account for correlations, and the latter one deals with collinearities of predictors in mechanistic ways, but is unable to detect the directionality and hierarchy of the relationship. We believe that the only way to examine the hierarchy of predictors explaining soil C would be a SEM-like technique, but not enough is known about the appropriate hierarchy and causal relationships between the predictors to build a hypothetical SEM model which would not be over-speculative. Therefore, we prefer to keep the original question of our manuscript, clarifying the question and the implications of the answer, and reducing the claims about causality of the detected relationship between soil C and dominance of EcM plants.

R.3.3. This is especially important as data presented in Table 1 seems to indicate that mycorrhizal type is ancillary to biome in predicting soil C storage in all cases.

Re: Indeed mycorrhizal type factor explains less variance than biome factor. This is expected, based on the body of literature on the impacts of soil, climate and landcover types on soil C. Our analysis is unique in showing that on top of these impacts, dominance of EcM vegetation (i.e. larger biomass of EcM plants in an ecosystem) is associated with higher soil C across all biomes, while for AM such relationship is not found.

R.3.4. Uncertainty. From the map in Supplemental Figure 2, it seems like many areas of the western hemisphere, central Asia, and Africa are poorly validated in this dataset. This is not surprising because these are often the areas with lower sampling effort. However, this may create uncertainty/error in your models. Therefore, another map of this uncertainty should be generated for all of the ecosystem C estimates that you present here. While this may show that we don't know much about the constraints of C in these areas, it is also an opportunity to point out future research sampling efforts. This should be able to replace the current Figure 2, which is already covered in the text.

Re: We agree that there is less data available in Central Asia and Africa than in other regions. We added a discussion on this point in the main text [lines 141-144]. However, generation of an uncertainty map for this specific area would not be possible, because our validation procedure is independent of the map creation procedure, i.e. our maps are not a result of modelling, but a result of the data assembly process [lines 479-492, Supplementary Tables 5-8]. However, we have included a note stating that map data for tropical areas, especially for those in Asia and Africa, should be considered with caution (lines 142-143), and that more sampling efforts are needed for this area (lines 143-144).

R3.5. Lns 322-324 suggest that AM fungi are not functional in agricultural systems. This is inaccurate

as many studies have surveyed the richness, composition, and functioning of AM fungi in these systems (e.g., Verbruggen et al. 2015 Soil Biology and Biochemistry).

Thank you very much for this inquiry. We agree that our current statements on the mycorrhizal associations of crops have been indeed formulated unclearly. In fact we do consider the majority of crops, except those that belong to the *Brassicaceae* family, to be arbuscular mycorrhizal. This is reflected in the Supplementary Tables 5-8. We have corrected this through the text (lines 171-172; 367-376). The main outcome of the analysis of the agricultural impacts is the quantification of the loss of ectomycorrhizal vegetation which is replaced by AM and some NM plants within the agricultural land conversions. This is reported in lines 187-189.

R3.6. At the very least other global change drivers (e.g., N deposition, climate change, etc.) should be compared to land-use change effects on these distributions. I suspect that an accurate accounting of AM fungi in agricultural systems may lessen many of these findings or fall within the range of uncertainty for the other global maps.

Re: We agree with the former part of this suggestion. Also Reviewer 2 has raised the same point. Please see our reply and the related manuscript changes under R2.9, R2.10, R2.11. Concerning the accurate accounting for AM in the agricultural system – we in fact do account for them, but this has been unclearly described in the manuscript. Please see our response to R 3.4, and the Supplementary Tables 6 and 8.

R3.7. Lns 414 – R scripts should be available in the manuscript as a supplement.

Re: we provide the set of R scripts used for map assembly, in a public GitHub repository https://github.com/nasoudzilovskaia/Soudzilovskaia_NatureComm_MycoMaps

Reviewers' comments:

Reviewer #1 (Remarks to the Author):

I thank the authors for their work on the revised analysis. Most of my major criticisms are sufficiently addressed. I do have some remaining recommendations and concerns. I believe these findings will substantially advance the field, and enable other global ecologists and biogeochemists to place their findings in a mycorrhizal context.

1. I am still confused regarding the accuracy of the validation (R1.3 in the response to review). Authors now report RMSD, and discount R2 as a useful metric because R2 doesn't reflect how well data fit the 1:1 line. This is fine, and authors can calculate R2 relative to a 1:1 line, rather than to the best fit line. However, what is really missing is a visualization of the fits. If authors could plot fitted vs. observed, as well as the 1:1 line, any reader could get a handle on how well the models capture the variation, and this is also where authors can report RMSD statistics (or any other goodness of fit metric). This data visualization is important, as these relationships are the foundation of your spatial scaling.

2. I'm apprehensive regarding authors desire to publish the mycorrhizal list as a separate paper. This list is foundational to the results of this paper (the analysis would not be possible without it). If authors can confirm that this list is now submitted, and that the final list will be made freely available (open access, no strings attached) in association with the up-coming New Phytologist publication they mention, I would be far less concerned.

Reviewer #2 (Remarks to the Author):

First, my apologies for delaying the review process. Due to unforeseen circumstances I took longer to complete the review than normal.

The manuscript has been improved in many ways, and the responses to previous reviews were thorough in most respects. Overall, I still think this makes a potentially valuable contribution. However, I still have concerns, not least about the estimation of errors.

It is good to see attempts to include uncertainty measures for the maps, but these seem to have some serious shortcomings:

(i) The uncertainties seem to be simply averaged and that average value given. That is of very limited use, for two main reasons.

First, the uncertainties should be mapped and provided as another layer of information, so that each estimated value is provided alongside its associated uncertainty measure. This does not seem to have been done (at least, I did not find it or any mention of it).

Second, average values for errors are only meaningful if they apply in a linear way to the estimates. But for the % values (see Fig. 1) this is not the case. Many of the estimates are near 0 or near 1, and in these cases there is a hard boundary in one direction on the possible error, meaning that the errors are reduced near these boundaries. Similarly, a value of 1.5% +/- 11% is not meaningful.

(ii) The uncertainties seem to have been artificially reduced by the authors arbitrarily removing a large number of the largest errors. This is described in lines 546-7, 'As RMSD is known to be highly prone to outliers, these were excluded from the calculations at probability level 0.90'.

First, this is not clear. I do not know what '[outliers] were excluded from the calculations at probability level 0.90' means. Clearly it is removing the data with largest errors (a large number of such data), but I can only guess at the details.

Second, RMSD is indeed sensitive to large values because it is based on squared errors, therefore emphasising large errors. But that is the standard way of measuring variation (variance, standard deviation, etc). It is NOT standard to greatly reduce that variation before measuring it! If there are genuine outliers then there may be a case for removing them, but only after careful consideration on a case-by-case basis. Blanket removal of all the data points with the biggest errors - the ones that would increase the error estimates the most - is simply not appropriate in my opinion, and will have seriously biased the error estimate. Specifically, it will have greatly reduced the estimates of error/uncertainty, making the data in the maps appear much more accurate than they actually are. In my opinion, this MUST BE ADDRESSED BEFORE ACCEPTANCE, and fixed. Note: the authors should not fear relatively high error estimates; high error estimates are surely appropriate for data like this, and I would be suspicious of low ones. Further, the data are valuable – the more so with appropriate error estimates.

(iii) There is confusion over the terminology. The error estimates in several places are referred to as 'accuracy', which is simply wrong. This produces, for example, the statement on line 140 that 'The averaged accuracy of our maps is estimated at 8-15%.' At least I assume this is simply an error in using the word 'accuracy'; if the accuracy of the maps is actually estimated at only 8-15% then this paper should be immediately rejected as having no meaningful data!

(iv) I could find no estimate of the uncertainty or error associated with Fig. 3. Also, the error estimates for Fig.2 are not given in the caption.

Other issues include:

It is disappointing that the mycorrhizal associate dataset is not being published here, but instead in another paper. This diminishes the value of the current manuscript.

On the positive side, if I have understood correctly the data underlying the maps and graphs in the current manuscript are being released via Github. That is important, though I suggest that a more formal, stable repository would be better (can be in addition to Github).

The statistical analysis of the relationship between soil carbon content and biomass distribution of mycorrhizal types seems problematic. There are considerable errors on both axes of this regression, so standard regression (which assumes no errors on X) is not appropriate. There is also uncertainty about the direction of causality (see rebuttal). Both these suggest that model II regression should be used.

Lines 245-7 say 'The correlation between the ISRIC-WISE predicted soil carbon and the original data that has used to generate the ISRIC soil map is within the range of 40-60%'. This is unclear. Correlations are measured using correlation coefficients, and they do not have units of %.

Rebuttal point R.2.9.

My original comment: 'In the expected fractions before human influence, should I be concerned that assumptions are made about eg depth distribution of mycorrhizas in the soil, when we only have data from a time when enhanced global N deposition (among other anthropogenic phenomena) has been occurring for decades, potentially altering these patterns?'

Response: 'We agree that nitrogen deposition is another important factor that have caused shifts in distributions of mycorrhizal vegetation. [...]'

This is a disappointing response. Nitrogen deposition was just an example, hence the 'eg' in the text of my original comment. My comment was clearly a more general point. While the specific answer about N deposition is good, it does not address the wider issue.

Rebuttal point R.3.4.

'However, generation of an uncertainty map for this specific area would not be possible, because our validation procedure is independent of the map creation procedure is independent of the map creation procedure [...]'

This is also a disappointing response. I do not believe that it is 'not possible' to produce an uncertainty map for that area. Instead, I read this response as saying that the authors chose not to do the work required to make such a map via another means.

Reviewer #3 (Remarks to the Author):

The authors have addressed all of my previous comments.

Minor edits: Lines 479-492 have multiple typos.

Dear Editor, Dear Reviewers,

Herewith we would like to submit the revised version of our paper “Global mycorrhizal plant distribution linked to terrestrial carbon stocks” . We have addressed all reviewers’ comments. Below we provide the detailed point by point explanation of the changes introduced into the manuscript. Our replies to the reviewers, as well as the new/modified text in the manuscript are shown in green colour. In reply to the editor’s request about the status of the paper presenting the mycorrhizal dataset underlying our paper, we would like to inform you that the paper has been submitted to the New Phytologist journal and placed into the pre-print Biorxiv repository (<https://doi.org/10.1101/717488>). We attach the submitted version of the paper and its supplementary tables for your convenience. Please find more details in the reply to Reviewer 1 request R1.3.

Reviewer #1 (Remarks to the Author):

R1.1. I thank the authors for their work on the revised analysis. Most of my major criticisms are sufficiently addressed. I do have some remaining recommendations and concerns. I believe these findings will substantially advance the field, and enable other global ecologists and biogeochemists to place their findings in a mycorrhizal context.

Thank you very much for this positive evaluation.

R1.2. I am still confused regarding the accuracy of the validation (R1.3 in the response to review). Authors now report RMSD, and discount R2 as a useful metric because R2 doesn't reflect how well data fit the 1:1 line. This is fine, and authors can calculate R2 relative to a 1:1 line, rather than to the best fit line. However, what is really missing is a visualization of the fits. If authors could plot fitted vs. observed, as well as the 1:1 line, any reader could get a handle on how well the models capture the variation, and this is also where authors can report RMSD statistics (or any other goodness of fit metric). This data visualization is important, as these relationships are the foundation of your spatial scaling.

We agree with the reviewer’s arguments. We have made scatter plots of the validation data points vs map data. We show the scatter plots in Extended Data Figure 3. In the main text we refer to this figure in the lines 131-132: “*The relationship between the validation data and our estimates is shown in Extended Data Figure 3.*”

This figure replaces the previous figure showing histograms for individual validation datasets. Because many data points in the scatters overlap or are very close together, we additionally show the 2d kernel density distribution of the data. In order to show the deviation of the data from 1:1 line we calculated the mean averaged error (MAE) metric, which is a direct expression of that deviation. The MAE values are reported in the caption of the figure. We have removed the text explaining the RMSD calculations and have replaced it by the text reporting MAE calculations (lines 481-487 in the Methods section): “*In order to quantify the differences between the validation datasets and our maps, we have calculated the Mean Averaged Error (MAE) of the difference between our maps and validation datasets. MAE expresses the deviation between two spatial data sets from the 1:1 line. For AM, EcM and NM vegetation fractions MAE is 18.7%, for EcM it is 13.6%, for NM it is 4.7%, respectively. Due to a virtual absence of information about distribution of solely ErM vegetation, direct validation of the ErM maps was impossible.*”

R1.3. I'm apprehensive regarding authors desire to publish the mycorrhizal list as a separate paper. This list is foundational to the results of this paper (the analysis would not be possible without it). If authors can confirm that this list is now submitted, and that the final list will be made freely available (open access, no strings attached) in association with the up-coming New Phytologist publication they mention, I would be far less concerned.

The paper presenting the mycorrhizal plants database has been submitted to the New Phytologist journal. We have also placed the paper as a pre-print into the biorxiv pre-print depository (<https://doi.org/10.1101/717488>). To enable a smooth review process we attach the proof of the New Phytologist submission to this current submission of our paper. The data used in this paper is presented in the Supplementary Table 3. This table was used to assign mycorrhizal status to the dominant plant species within Bailey ecoregion * continent * land cover combinations during the process of map creation. Being a supplementary to the New Phytologist paper the table is freely open for use, without any restrictions besides a request to cite the New Phytologist paper. Moreover the table is already available in the pre-print (<https://doi.org/10.1101/717488>). In the new version of the manuscript we provide a reference to this paper (ref 30).

Reviewer #2 (Remarks to the Author):

R2.1. First, my apologies for delaying the review process. Due to unforeseen circumstances I took longer to complete the review than normal.

The manuscript has been improved in many ways, and the responses to previous reviews were thorough in most respects. Overall, I still think this makes a potentially valuable contribution.

Thank you very much for this positive evaluation!

R2.2. However, I still have concerns, not least about the estimation of errors.

It is good to see attempts to include uncertainty measures for the maps, but these seem to have some serious shortcomings:

(i) The uncertainties seem to be simply averaged and that average value given. That is of very limited use, for two main reasons. First, the uncertainties should be mapped and provided as another layer of information, so that each estimated value is provided alongside its associated uncertainty measure. This does not seem to have been done (at least, I did not find it or any mention of it). Second, average values for errors are only meaningful if they apply in a linear way to the estimates. But for the % values (see Fig. 1) this is not the case. Many of the estimates are near 0 or near 1, and in these cases there is a hard boundary in one direction on the possible error, meaning that the errors are reduced near these boundaries. Similarly, a value of 1.5% +/- 11% is not meaningful.

We agree with this point. In the new version of the manuscript we have calculated the spatially explicit uncertainties of our maps following a different algorithm allowing a comprehensive per pixel estimation of uncertainty (see below), and provided the new uncertainty maps as Extended data Figures 4 (uncertainties of current distribution of mycorrhizal vegetation), 6 (uncertainties of cropland-free distribution of mycorrhizal vegetation), and 7 (uncertainties of losses of mycorrhizal vegetation). We have added the algorithms applied hereto to the main text (lines 145-154), and to the Methods section (lines 505-523).

The main text: “We examined the uncertainty of our maps based on uncertainties of tree, shrubs and herbaceous plant biomass fractions within the land cover types (REF CCI, REFS), and the number of data sources used to assess mycorrhizal fractions of plant biomass within each combination of Bailey ecoregion x continent; see methods for details. Extended Data Figure 4 shows spatial distribution of map uncertainties. The mean uncertainties of AM, EM, ER, NM maps are 19.6, 17.6, 14.6, and 15.0% at the 90% confidence interval. Overall, tropical areas have the highest uncertainties of the mycorrhizal fraction data, reaching 50% (AM). in the Amazon region. Therefore, our maps should be used with caution for these areas. Future sampling efforts of mycorrhizal vegetation distribution should be more focussed on tropical areas of Asia, Africa and South America.”

Methods:” We quantified the uncertainty in our maps of mycorrhizal vegetation fractions by applying the error propagation rules to the formulas used to calculate the biomass fractions of mycorrhizal plants per grid cell. Hereto we used the data provided by (^{REF CCI land 2, and 36,63-65}) to estimate the uncertainty associated with relative biomass of trees, shrubs, and herbaceous vegetation in each CCI land cover class. The uncertainty in the proportion of each mycorrhizal type in a given Bailey x continent combination was set to $1/\sqrt{n}$, where $n=b+c/3+g/20$. In this formula b is a number of literature sources describing the vegetation composition in a given Bailey ecoregion x continent combination (Extended Data Table 3), while c and g are the numbers of literature sources describing vegetation composition at continent level and global levels, respectively. These latter sources were given less weight, because of their general orientation, though these sources were used only if they were providing information relevant for the combination of Bailey ecoregion * continent under consideration. This procedure was applied to the maps of the current distribution of mycorrhizal fractions (Figure 1) and for the distribution of mycorrhizal fractions in the cropland-free world (Extended Data Figure 5). The resulting maps of uncertainties are shown in Extended Data Figures 4 and 6, respectively. We calculated the uncertainties in the estimations of changes in biomass fractions of mycorrhizal vegetation induced by crop cultivation and pastures (Figure 3) by applying the error propagation rule for the mathematical operation of subtraction. The resulting uncertainty map is shown in Extended Data Figure 7.”

R2.3. The uncertainties seem to have been artificially reduced by the authors arbitrarily removing a large number of the largest errors. This is described in lines 546-7, ‘As RMSD is known to be highly prone to outliers, these were excluded from the calculations at probability level 0.90’.

First, this is not clear. I do not know what ‘[outliers] were excluded from the calculations at probability level 0.90’ means. Clearly it is removing the data with largest errors (a large number of such data), but I can only guess at the details.

Second, RMSD is indeed sensitive to large values because it is based on squared errors, therefore emphasising large errors. But that is the standard way of measuring variation (variance, standard deviation, etc). It is NOT standard to greatly reduce that variation before measuring it! If there are genuine outliers then there may be a case for removing them, but only after careful consideration on a case-by-case basis. Blanket removal of all the data points with the biggest errors - the ones that would increase the error estimates the most - is simply not appropriate in my opinion, and will have seriously biased the error estimate. Specifically, it will have greatly reduced the estimates of error/uncertainty, making the data in the maps appear much more accurate than they actually are. In my opinion, this MUST BE ADDRESSED BEFORE ACCEPTANCE, and fixed. Note: the authors should not fear relatively high error estimates; high error estimates are surely appropriate for data like

this, and I would be suspicious of low ones. Further, the data are valuable – the more so with appropriate error estimates.

In combination with R1.2, we decided to replace the RMSD calculations by calculations of mean averaged error (MAE). We calculated the MAE based on the full datasets, without excluding any outliers, and reported the outcomes in the caption of the new Extended Data Figure 3. In order to compare the MAE estimations to the estimations of RMSD, we have also calculated the latter based on the full dataset. The respective RMSD values are similar to MAE: MAE for AM, EcM, NM are 18.7, 13.6 and 4.7%; and the respective RMSDs are 22.3, 18.3, and 5.8%. Given that both estimates are widely accepted as a measure of correlation between spatial datasets, and in order to avoid repetition, we do not report the RMSD in the new version of the manuscript.

R2.4. There is confusion over the terminology. The error estimates in several places are referred to as ‘accuracy’, which is simply wrong. This produces, for example, the statement on line 140 that ‘The averaged accuracy of our maps is estimated at 8-15%.’ At least I assume this is simply an error in using the word ‘accuracy’; if the accuracy of the maps is actually estimated at only 8-15% then this paper should be immediately rejected as having no meaningful data!

Thank you for pointing out to this lack of clarity in the text. We have corrected the terminology throughout the manuscript, using exclusively the word “uncertainty”

R2.5. I could find no estimate of the uncertainty or error associated with Fig. 3.

We have added the Extended Data Figure 7 to show the uncertainties associated with Fig. 3.

R2.6. Also, the error estimates for Fig.2 are not given in the caption.

We have added the error estimates to the caption of the Fig. 2. These error estimates have been re-calculated using the new maps of uncertainties in mycorrhizal biomass fractions. The calculations have been clarified in the methods section (lines 536-539):

“To assess uncertainty of estimations of carbon storage in mycorrhizal vegetation we used the rule of uncertainty propagation through the operation of multiplication. Thereto we used the uncertainty of mycorrhizal biomass fraction data and the per biome uncertainty of biomass carbon data, as provided by Liu et al³⁸”

R2.7. Other issues include:

It is disappointing that the mycorrhizal associate dataset is not being published here, but instead in another paper. This diminishes the value of the current manuscript.

We understand this concern. However the mycorrhizal dataset is a result of an effort of a number of people who played a prominent role in assembly of the dataset but did not participate in the current work. Therefore we prefer to split these two publications. As discussed in the reply to R1.3. the paper presenting the mycorrhizal dataset has been submitted to the New Phytologist and placed into the preprint repository Biorxiv (<https://doi.org/10.1101/717488>). Therefore the data used to create the maps is fully available to the research community.

R2.8. On the positive side, if I have understood correctly the data underlying the maps and graphs in the current manuscript are being released via Github. That is important, though I suggest that a more formal, stable repository would be better (can be in addition to Github).

We agree with this suggestion. In addition to Github the data will be placed to the Netherlands governmental data repository DANS (Data Archiving and Networked Services (<https://dans.knaw.nl/en>)). We have added a sentence in the main text reporting this (lines 646-651): “All the codes and author’s data used to create the maps of mycorrhizal vegetation biomass, and all the data underlying Figures 1-4, Supplementary Figs 3-6, Table 1 and Supplementary Tables 9-11 are available at a GitHub repository https://github.com/nasoudzilovskaia/Soudzilovskaia_NatureComm_MycoMaps and at the Netherlands governmental data repository DANS (www.dans.nl)”

R2.9. The statistical analysis of the relationship between soil carbon content and biomass distribution of mycorrhizal types seems problematic. There are considerable errors on both axes of this regression, so standard regression (which assumes no errors on X) is not appropriate. There is also uncertainty about the direction of causality (see rebuttal). Both these suggest that model II regression should be used.

We thank the reviewer for sharing his/her concerns in relation to this analysis. However, we do not agree that a model II regression (i.e. standardized major axes regression) is the best solution here. An important assumption of the model II regression is that values of the x and y axes have a bivariate normal distribution (see for instance the paper of Pierre Legendre “Model II regression user’s guide, R edition”, available at <https://cran.r-project.org/web/packages/lmodel2/vignettes/mod2user.pdf>), which is not a case for our data. In our case, the data of the x axis has a binomial distribution. Therefore we originally opted for the glm analysis of the Gaussian family. Despite not fulfilling the assumptions of a model II regression, we considered the suggestion of the reviewer and additionally ran the standardized major axes (SMA) analysis (model II regression) on our data. The results of this analysis turn out to provide exactly the same conclusions, as our initial analysis, i.e. stressing the importance of EcM plant biomass fraction as a positive predictor of soil C. We found that the positive relationship between EcM biomass fraction and soil C was hold across all biomes, while the intercept values were biome-dependent, indicating importance of biome as a second predictor. The relationship to AM plant biomass fraction is biome-dependent and, though the relationship is generally negative, the slope values vary strongly among biomes, ranging from positive to negative values. This suggests an idiosyncratic relationship between soil C and AM plant biomass fraction, with soil C being strongly affected by interactions between biome factor and AM plant biomass. This is exactly the same conclusion that we have derived from our initial analysis. The table below shows the summary of the SMA analysis.

However, next to not fulfilling the assumptions, the SMA method has another serious shortcoming: according to the best of our knowledge it does not allow the estimation of the relative importance of distinct predictors (i.e. biome vs mycorrhizal type abundances), which is an essential part of our analysis. Given that limitation, and considering the violation of the bivariate normality assumption, we prefer to keep the original report on the outcomes of the glm analysis in the manuscript.

Table 1. Outcomes of the Model II regression (SMA) analysis on the relationship between soil C and EM biomass fraction.

Response Variable	Predictor	Slope	P-value slope	R ²
C in top soil (0-20 cm)	EM	0.309	<0.001	0.41
	EM and biome	Varies per biome, but is always positive		
	AM	-0.275	<0.001	0.39
	EM and biome	Varies per biome, having ranges from positive to negative		
C at 20-60 cm soil depth	EM	0.623	<0.001	0.24
	EM and biome	Varies per biome, but is always positive		
	AM	-0.556	<0.001	0.19
	EM and biome	Varies per biome, having ranges from positive to negative		
C at 60-100cm soil depth	EM	0.57	<0.001	0.21
	EM and biome	Varies per biome, but is always positive		
	AM	-0.509	<0.001	0.19
	EM and biome	Varies per biome, having ranges from positive to negative		

R2.10. Lines 245-7 say ‘The correlation between the ISRIC-WISE predicted soil carbon and the original data that has used to generate the ISRIC soil map is within the range of 40-60%’. This is unclear. Correlations are measured using correlation coefficients, and they do not have units of %.

We have re-phrased this sentence as “*Analyses of relationships between ISRIC-WISE predicted soil carbon and the original data that were used to generate the ISRIC soil map yield R²-values in the range of 0.4-0.6*” (lines 244-246)

R2.11. Rebuttal point R.2.9.

My original comment: ‘In the expected fractions before human influence, should I be concerned that assumptions are made about eg depth distribution of mycorrhizas in the soil, when we only have data from a time when enhanced global N deposition (among other anthropogenic phenomena) has been occurring for decades, potentially altering these patterns?’

Response: ‘We agree that nitrogen deposition is another important factor that have caused shifts in distributions of mycorrhizal vegetation. [...]’

This is a disappointing response. Nitrogen deposition was just an example, hence the ‘eg’ in the text of my original comment. My comment was clearly a more general point. While the specific answer about N deposition is good, it does not address the wider issue.

We agree that it is necessary to discuss the assumptions made in the analysis of human impacts on the distribution patterns of mycorrhizal vegetation. We have added the following text to the manuscript (lines 278-284): *“Analyses of agricultural impacts presented in this paper are based on the assumption that these impacts are limited to shifts in plant species composition, and do not encompass shifts in soil water and nutrient availability, which could affect activity of mycorrhizal fungi, depth distribution of mycorrhizas in the soil, and shifts among mycorrhizal fungal species composition due to an introduction of exotic species. While such simplifications are necessary for the analyses reported in this paper, they should be considered when interpreting our results.”*

R2.12. Rebuttal point R.3.4.

‘However, generation of an uncertainty map for this specific area would not be possible, because our validation procedure is independent of the map creation procedure is independent of the map creation procedure [...]

This is also a disappointing response. I do not believe that it is ‘not possible’ to produce an uncertainty map for that area. Instead, I read this response as saying that the authors chose not to do the work required to make such a map via another means.

In the new manuscript version we provide global maps of uncertainties, see rebuttal point R2.2.

Reviewer #3 (Remarks to the Author):

R3.1. The authors have addressed all of my previous comments.
Minor edits: Lines 479-492 have multiple typos.

Thank you very much for the careful check. We have corrected the typos.

REVIEWERS' COMMENTS:

Reviewer #1 (Remarks to the Author):

The authors have addressed all of my major concerns, though I do still think should report an R² value (relative to the 1:1 line) for Extended Data Figure 3. This is common practice in evaluating model accuracy in global scale analyses such as this. The kernel density is still hard to interpret.

The analysis here represents a breakthrough in mapping a major aspect of the forest microbiome, and linking the forest microbiome to the carbon sequestration capacity of the Earth. The work has direct implications for conservation strategies that seek to manage soil carbon. I believe this work will be well received by a huge breadth of researchers, who are increasingly realizing that mycorrhizal effects pervade all aspects of forest ecology and function. I am very supportive of this paper.

Reviewer #2 (Remarks to the Author):

The authors have taken the previous concerns seriously and made appropriate changes and/or replied convincingly, in my opinion. I have no further substantial criticisms. I think this should be an important contribution to the literature.

On the specific point of model II regression, I am glad that the authors ran such regressions and reassured that the results are qualitatively the same. I agree with the authors' approach of focusing the reporting on the glm results (similarly the treatment of spatial autocorrelation). My one suggestion for a minor revision (at the editor's discretion) is that this modelling exercise is reported in the manuscript in a similar way to the one for spatial autocorrelation (Section 'Dealing with spatial autocorrelations', lines 608-636).

The manuscript does need a good copy-edit: there are numerous minor issues of grammar, punctuation, inconsistency, citation formatting, etc. I trust the journal will take care of this copy-edit.

The copy-editing should include the supplementary material. Particular attention should be paid to the captions of the Extended Data Figures and Tables. Many of these are not as clear as they should be and/or lack information such as what the abbreviations mean; a small amount of work on improving these captions will greatly enhance the supporting information for this paper. This is particularly important given the potential for this paper to become well cited and the data widely used.

I have reviewed each of the submissions of this manuscript, and would like to say that the authors have on the whole responded well to the criticisms throughout the review process (both my criticisms and those of the other referees). The result is a much improved paper that has a concomitantly improved chance of having a major impact on the field.

Signed: Richard Field.

Dear Editor, Dear Reviewers,

We appreciate the contributions by the reviewers and editor to improve our manuscript “*Global mycorrhizal plant distribution linked to terrestrial carbon stocks*”, which we here resubmit for your final review. We have addressed all reviewers’ comments. Below we provide the detailed point by point explanation of the changes introduced in the manuscript in response to the Referees’ comments. Our replies to the reviewers are shown in blue colour. As required by the Editor, we submit a separate file including a point-by-point response to editorial suggestions.

Reviewer #1 (Remarks to the Author):

The authors have addressed all of my major concerns, though I do still think should report an R² value (relative to the 1:1 line) for Extended Data Figure 3. This is common practice in evaluating model accuracy in global scale analyses such as this. The kernel density is still hard to interpret.

Re: Thank you for acknowledging our efforts. We have added the value of R² to 1:1 line to the caption of the figure presenting the validation results (Supplementary Figure 3): “R² to 1:1 line is 0.81, 0.92 and 0.93 for AM, EcM and NM biomass vegetation fractions.”

The analysis here represents a breakthrough in mapping a major aspect of the forest microbiome, and linking the forest microbiome to the carbon sequestration capacity of the Earth. The work has direct implications for conservation strategies that seek to manage soil carbon. I believe this work will be well received by a huge breadth of researchers, who are increasingly realizing that mycorrhizal effects pervade all aspects of forest ecology and function. I am very supportive of this paper.

Re: Thank you very much for this highly positive evaluation and sharing our enthusiasm about the implications of our work.

Reviewer #2 (Remarks to the Author):

The authors have taken the previous concerns seriously and made appropriate changes and/or replied convincingly, in my opinion. I have no further substantial criticisms. I think this should be an important contribution to the literature.

Re: Thank you for acknowledging our efforts

On the specific point of model II regression, I am glad that the authors ran such regressions and reassured that the results are qualitatively the same. I agree with the authors’ approach of focusing the reporting on the glm results (similarly the treatment of spatial autocorrelation). My one suggestion for a minor revision (at the editor’s discretion) is that this modelling exercise is reported in the manuscript in a similar way to the one for spatial autocorrelation (Section ‘Dealing with spatial autocorrelations’, lines 608-636).

Re: We have incorporated this useful suggestion in Supplementary Table 4. The text referring to this table has been added in the lines 623-625: “As this analysis encompasses considerable errors on both axis of the regression, we have additionally checked if a model II regression would yield qualitatively similar results, confirming this was the case (Supplementary Table 4).”

The manuscript does need a good copy-edit: there are numerous minor issues of grammar, punctuation, inconsistency, citation formatting, etc. I trust the journal will take care of this copy-edit. The copy-editing should include the supplementary material. Particular attention should be paid to the captions of the Extended Data Figures and Tables. Many of these are not as clear as they should be and/or lack information such as what the abbreviations mean; a small amount of work on improving these captions will greatly enhance the supporting information for this paper. This is particularly important given the potential for this paper to become well cited and the data widely used.

Re: We appreciate the comment. The manuscript has been checked and corrected by native speaker co-authors. We have also clarified captions in the Supplementary materials.

I have reviewed each of the submissions of this manuscript, and would like to say that the authors have on the whole responded well to the criticisms throughout the review process (both my criticisms and those of the other referees). The result is a much improved paper that has a concomitantly improved chance of having a major impact on the field.

Signed: Richard Field.

Re: Dear Richard, thank you very much for the constructive comments and for acknowledgment of our efforts to address them.